# SimulRAG: Simulator-based RAG for Grounding LLMs in Long-form Scientific QA

## Abstract

Large language models (LLMs) show promise in solving scientific problems. They can help generate long-form answers for scientific questions, which are crucial for comprehensive understanding of complex phenomena that require detailed explanations spanning multiple interconnected concepts and evidence. However, LLMs often suffer from hallucination, especially in the challenging task of long-form scientific question answering. Retrieval-Augmented Generation (RAG) approaches can ground LLMs by incorporating external knowledge sources to improve trustworthiness. In this context, scientific simulators, which play a vital role in validating hypotheses, offer a particularly promising retrieval source to mitigate hallucination and enhance answer factuality. However, existing RAG approaches cannot be directly applied for scientific simulation-based retrieval due to two fundamental challenges: how to retrieve from scientific simulators, and how to efficiently verify and update long-form answers. To overcome these challenges, we propose the simulator-based RAG framework (SimulRAG) and provide a long-form scientific QA benchmark covering climate science, epidemiology, and urban planning with ground truth verified by both simulations and human annotators. In this framework, we propose a generalized simulator retrieval interface to transform between textual and numerical modalities. We further design a claim-level generation method that utilizes uncertainty estimation scores and simulator boundary assessment (UE+SBA) to efficiently verify and update claims. Extensive experiments demonstrate SimulRAG outperforms traditional RAG baselines by 30.4% in informativeness and 16.3% in factuality. UE+SBA further improves efficiency and quality for claim-level generation.

## 1 Introduction

The lofty goal of developing AI scientists has driven extensive LLM research across various scientific tasks, ranging from exam-style question answering (QA) (Lu et al., 2022; Zhang et al., 2024) to hypothesis proposal (Wang et al., 2024; Yang et al., 2024) and experiment design (Chen et al., 2024; Mialon et al., 2023). Among these, long-form QA is an important task that requires AI scientists to provide answers that blend multiple scientific claims. This task tests an AI scientist's ability to reason through complex scientific phenomena and provide comprehensive explanations from different perspectives (Rein et al., 2024; Lee et al., 2023). For example, in epidemiology, predicting disease spread dynamics requires analyzing multiple interconnected factors—including transmissibility rates, incubation periods, clinical severity, seasonal variations, population demographics, and contact mixing patterns—to model how diseases propagate through communities over time (Chang et al., 2020; Cramer et al., 2022). However, comprehensive studies on LLMs for long-form scientific QA are limited, and existing works on general free-form QA already manifest persistent hallucination issues (Farquhar et al., 2024).

Recent works have shown that grounding LLMs with external knowledge sources can help mitigate hallucination and improve answer factuality (Schick et al., 2023; Patil et al., 2024). In the scientific domain, it is natural to consider scientific simulators or corresponding emulators as tools to solve scientific problems (Ren et al., 2025; Ma et al., 2024). Compared with static textual knowledge bases such as literature reviews, querying simulators can capture evolving dynamics and provide more detailed information about specific scientific phenomena, making simulators more informative

Figure 1: Left: Overall SimulRAG structure, including the simulator-based retriever and claim-level generator. Right: Simulator retrieval interface: (1) prompting LLM with question and handbook to extract simulator parameter settings; (2) executing simulator with parameters to obtain simulation outputs; (3) converting outputs to textual context via predefined or LLM-generated templates.

tools to use and in a more timely fashion (Ren et al., 2025). Existing works have focused on fine-tuning LLMs to use scientific tools for predefined tasks (Lyu et al., 2024; Thulke et al., 2024; Zhang et al., 2024), which requires expensive computational resources, predefined question templates, pre-collected datasets, and is therefore not generalizable and scalable across different scientific domains.

Retrieval-Augmented Generation (RAG) approaches have shown promise for enhancing LLMs by incorporating external knowledge sources to improve answer factuality and informativeness (Lewis et al., 2020b; Fan et al., 2024; Gao et al., 2023) without additional fine-tuning, making them promising candidates for grounding LLMs as trustworthy AI scientists. RAG consist of two main components: a *retriever* that searches relevant documents and a *generator* that produces answers based on the retrieved content. Unfortunately, two fundamental challenges hinder the application of the existing RAG approaches to the simulation-based retrieval for long-form QA in scientific domains. First, the discrepancy between the textual space and the numerical space—where the simulation parameters and outputs reside in—presents challenges to querying and retrieving from scientific simulators. Second, existing RAG generators cannot effectively update long-form answers with new context due to the lack of fine-grained control.

To overcome these challenges, we introduce the simulator-based RAG framework (SimulRAG) for long-form scientific QA. The overall structure is illustrated in Figure 1 left. It provides a generalized simulator retrieval interface to transform between textual and numerical modalities, enabling seamless integration of scientific simulators into RAG systems. We introduce a granular generation method which decomposes the long-form answers into atomic claims then verifies and updates each claim. To further improve generation efficiency, we utilize uncertainty estimation scores and simulator boundary assessment (UE+SBA) to only verify and update claims when necessary. To systematically evaluate various methods for long-form scientific QA, we additionally construct the benchmark for such tasks using simulators as retrieval tools, covering climate modeling epidemiology and urban planning with ground truth verified by both simulations and human annotators to ensure high quality. The extensive experimental results demonstrate that SimulRAG outperforms traditional RAG baselines in factuality and informativeness, while UE+SBA improves efficiency and quality for claim-level generation.

Our contributions are summarized as follows:

- We introduce SimulRAG, a simulator-based RAG framework for long-form scientific QA.
- We propose a generalized simulator retrieval interface to transform between textual and numerical modalities.
- We present a claim-level generation method to improve long-form answer quality.
- We utilize uncertainty estimation scores and simulator boundary assessment (UE+SBA) to efficiently verify and update claims.
- We construct a long-form scientific QA benchmark for climate science and epidemiology.
- We conduct extensive experiments to verify SimulRAG framework and UE+SBA method effectiveness. Results show SimulRAG outperforms RAG baselines by 30.4% in informativeness and 16.3% in factuality.

## 2 RELATED WORK

### 2.1 SCIENTIFIC QUESTION ANSWERING

Scientific question answering encompasses multiple formats and domains. ScienceQA (Lu et al., 2022) contains multimodal multiple choice questions from high school textbooks. Microvqa (Burgess et al., 2025) provides multimodal reasoning benchmarks for microscopy-based research. These works focus on multi-choice questions (MCQs) rather than free-form QA. Climate Crisis QA (Zhu & Tiwari, 2023) and SciQAG-24D (Wan et al., 2024) explore synthetic data generation using LLMs, but suffer from hallucinations and lack scientific validity. CLIMAQA (Manivannan et al., 2024) provides automated evaluation frameworks for climate science QA. SciQA (Auer et al., 2023) benchmarks scientific QA using hand-crafted queries on the Open Research Knowledge Graph (Jaradeh et al., 2019). Both support free-form QA but address short-form questions. Their answers are simple, typically containing single claim. Our questions require multiple claims forming complete answers. We provide the benchmark for long-form scientific QA across climate modeling and epidemiological modeling domains. The ground truth answers are verified by both scientific simulators and human annotators to ensure high quality.

### 2.2 RETRIEVAL-AUGMENTED GENERATION (RAG) FOR LLMs

RAG systems consist of retrieval and generation components (Lewis et al., 2020b; Fan et al., 2024; Gao et al., 2023). The retriever searches relevant documents from knowledge bases by measuring distance between queries and documents. Two main retrieval methods exist: dense retrieval maps queries and knowledge into vector spaces (Khandelwal et al., 2019; Karpukhin et al., 2020; Lewis et al., 2020a), while sparse retrieval uses word-based matching (Robertson et al., 2009; Sparck Jones, 1972). The generation module takes queries and retrieved documents as input to generate final answers. Three generation strategies exist: input-layer integration combines queries with retrieved documents as LLM input (Ram et al., 2023; Izacard & Grave, 2020); output-layer integration refines generation results using retrieval (Khandelwal et al., 2019; Yogatama et al., 2021; Yu et al., 2023); intermediate-layer integration uses semi-parametric modules within generation models (Borgeaud et al., 2022; Wu et al., 2022). However, intermediate approaches require model access which is unavailable using LLM APIs (Ma et al., 2023). Existing RAG methods rely on search-based retrieval mechanisms unsuitable for scientific simulators. Recent work explored scientific tool integration (Lyu et al., 2024; Wang et al., 2024; Yang et al., 2024; Majumder et al., 2024), but targets predefined tasks rather than free-form scientific QA with versatile open-ended questions. Our work proposes the general RAG framework using simulators as retrieval tools.

### 2.3 CLAIM-LEVEL UNCERTAINTY ESTIMATION

Long-form answer uncertainty estimation decomposes answers into multiple claims for granular assessment. Several methods (Duan et al., 2023; Band et al., 2024) obtain claim uncertainty scores from long-form outputs but require white-box model access, limiting applicability to API-based LLMs. SelfCheckGPT (Manakul et al., 2023) extends self-consistency (Wang et al., 2022) to sentence-level uncertainty within long-form outputs, applicable to black-box LLMs. Mohri et al. (Mohri & Hashimoto, 2024) perform claim-level uncertainty estimation using conformal prediction. Jiang et al. (Jiang et al., 2024) improve granular uncertainty estimation through entailment graphs capturing fine-grained semantic information. We adapt this claim-level uncertainty method to guide the claim-level generation process, improving its efficiency and quality.

## 3 METHODOLOGY

### 3.1 OVERALL FRAMEWORK

Our proposed simulator-based RAG (SimulRAG) framework retrieves from scientific simulators to ground LLMs for long-form scientific QA tasks. The problem formulation is as follows. Given a open-ended question $q$, the SimulRAG framework first retrieves relevant simulation outputs $d$ from the scientific simulator $S$ through the simulator retrieval interface $I$:

$$d = I(S, q) \tag{1}$$

The output $d$ is converted to textual context $c$. The generator $G$ takes the question $q$, context $c$, and LLM model $M$ as input to generate the claim set $\mathcal{C} = \{c_1, c_2, \ldots, c_k\}$:

$$\mathcal{C} = G(q, c, M) \tag{2}$$

where the claim set $\mathcal{C}$ contains multiple atomic claims that form the final long-form answer $a$.

The overall SimulRAG framework is illustrated in Figure 1 left. Our main contributions in this framework are the design of simulator retrieval interface $I$ and the claim-level generator $G$. For the simulator retrieval interface $I$, we design a generalized approach to transform between textual and numerical modalities. For the claim-level generator $G$, we utilize uncertainty estimation scores and simulator boundary assessment (UE+SBA) to efficiently verify and update claims. The details of $I$ and $G$ are described in the following sections. The complete algorithm is presented in Algorithm 1.

### 3.2 SIMULATOR RETRIEVAL INTERFACE

Our simulator retrieval interface transforms between textual and numerical modalities. Figure 1 right illustrates the overall process. We use question $q$ and simulator $S$'s handbook as context $h$ to guide the LLM in understanding simulator functionality and parameter space. The LLM extracts multiple relevant parameter settings from question $q$ and context $h$. These parameters are transformed to JSON format and executed by simulator $S$ to obtain outputs $d$. The simulation outputs are converted to textual format $c$ using predefined templates for subsequent claim verification and updating. We can predefine output templates since simulator output format is fixed. The key challenge solved is extracting correct parameter settings from versatile open-ended questions. Unlike existing methods requiring predefined templates with token indicators, our method supports different types of open-ended questions. Prompts and templates used are provided in Appendix A.2.

### 3.3 CLAIM-LEVEL GENERATION

Traditional RAG generation methods directly produce answers given questions and retrieved context. This one-step mechanism often yields suboptimal informativeness and factuality. To improve this, we generate multiple diverse answers $\mathcal{A} = \{a_1, a_2, \ldots, a_m\}$ through $m$ LLM queries, covering different aspects of the question. Each answer $a_i$ is decomposed into atomic claims $\{c_{i1}, c_{i2}, \ldots, c_{in}\}$ following (Min et al., 2023), where each claim $c_{ij}$ represents an independently verifiable factual statement. Claims from different answers merge into a single deduplicated set $\mathcal{C} = \{c_1, c_2, \ldots, c_k\}$ using the LLM deduplication approach from (Jiang et al., 2024).

This claim-level decomposition serves two critical purposes: (1) enabling targeted verification and updates of individual claims rather than holistic response modification, providing more precise and flexible long-form answer refinement; (2) simplifying the verification task by focusing on atomic factual statements. Each claim represents a concise scientific assertion about phenomena, relationships, or quantitative predictions that can be directly validated against simulation outputs.

However, verifying all $k$ claims requires $O(k)$ simulator queries, creating computational bottlenecks. We address this through uncertainty estimation scores and simulator boundary assessment (UE+SBA), which selectively verify only uncertain and verifiable claims. Figure 2 illustrates the complete claim-level generation process.

**Claim-level Uncertainty Estimation** To assess the uncertainty estimation score of each claim $c_i$, we construct bipartite graphs between the answer set $\mathcal{A}$ and the claim set $\mathcal{C}$. Each node represents either an answer or a claim, while edges capture semantic entailment relationships between them. We estimate claim-level uncertainty using graph centrality metrics, specifically adopting closeness centrality which measures how close a claim node is to all other nodes.

$$\text{conf}(c_i) = \frac{|\mathcal{V}| - 1}{\sum_{u \in \mathcal{V}} d(c_i, u)} \cdot \frac{|\mathcal{V}|}{|\mathcal{V}_{c_i}|} \tag{3}$$

where $\mathcal{V}$ is the set of all nodes, $d(c_i, u)$ is the shortest path distance between claim node $c_i$ and node $u$, and $|\mathcal{V}_{c_i}|$ represents the size of the connected component containing $c_i$. Higher closeness

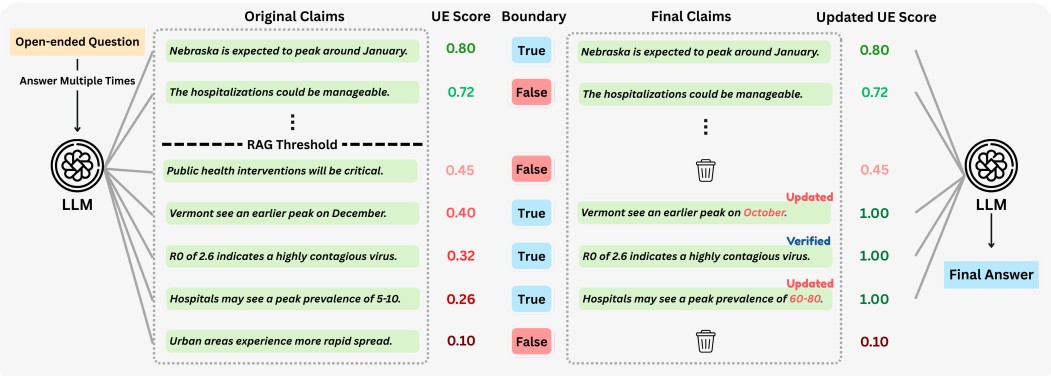

Figure 2: Claim-level generation process: (1) decompose long-form answers into atomic claims; (2) apply uncertainty estimation and simulator boundary assessment to select claims for verification; (3) update selected claims using simulation context; (4) integrate verified claims into a coherent answer.

centrality indicates a more central claim with stronger support from multiple answers, thus higher confidence. Our approach differs from (Jiang et al., 2024) in purpose: while they detect uncertain claims for abstention, we identify uncertain claims for verification and modification through simulation output context to improve answer factuality. We evaluate additional graph centrality metrics as uncertainty estimators, with results presented in Section 4.3.

**Simulator Boundary Assessment** Another important factor to consider is whether claim $c_i$ can be verified by simulator $S$. To this end, we introduce a boundary compatibility function

$$\text{bound}(c_i, h) \rightarrow \{0, 1\} \tag{4}$$

This function returns a binary value indicating whether claim $c_i$ parameters and conditions fall within simulator $S$ operational boundaries. We employ GPT-4o as an LLM judge, providing simulator handbook $h$ and claim $c_i$ for evaluation. The judge determines compatibility between claim parameters and simulator capabilities, returning 1 for compatible claims and 0 for incompatible ones. This assessment filters incompatible claims before verification, reducing unnecessary queries and improving efficiency.

**UE+SBA Selection and Verification and Updating.** We combine uncertainty estimation and boundary assessment for selective claim verification. A claim $c_i$ undergoes simulator verification when meeting two criteria: (1) uncertainty: $\text{conf}(c_i) < \tau$ where $\tau$ represents a predefined confidence threshold; (2) boundary compatibility: $\text{bound}(c_i, h) = 1$. This dual-criterion approach optimally allocates computational resources to uncertain yet verifiable claims.

Selected claims undergo verification using simulation context $c$ through three scenarios: (1) alignment: claims consistent with $c$ retain original content with $\text{conf}(c_i) = 1$; (2) contradiction: claims conflicting with $c$ are updated based on simulation evidence with $\text{conf}(c_i) = 1$; (3) indeterminate: unverifiable claims preserve original content and confidence scores. Finally, we apply confidence threshold $\kappa$ to filter low-confidence claims. The high-confidence set $\{c_i \in \mathcal{C} | \text{conf}(c_i) \geq \kappa\}$ integrates into the final answer $a'$.

### 3.4 SCIENTIFIC BENCHMARK GENERATION

We construct a benchmark dataset for long-form scientific QA using simulators as retrieval tools, covering climate modeling and epidemiology domains. For climate modeling, we utilize the climate emulator (Niu et al., 2024) trained on Coupled Model Intercomparison Project Phase 6 (Eyring et al., 2016) simulations, comprising general circulation and Earth system models representing the scientific standard for climate projection. This emulator enables efficient exploration of climate scenarios prohibitively expensive with full CMIP6 models. Focusing on four greenhouse and aerosol gases

| Climate | Epidemiology |
|---|---|
| **Q: Considering the climate projection for Syktyvkar, what is the total temperature change expected between the historical average in 2006 and the predicted temperature in 2040 under the ssp126 scenario, given a 20.41% increase in CO2 and a 46.77% increase in CH4 emissions? Does this change represent a warming or cooling trend, and is its magnitude considered negligible (less than 0.2°C), modest (0.2°C to 1.0°C), or significant (over 1.0°C)?**

**A: The total predicted temperature change is 0.29°C, which indicates a warming trend for Syktyvkar between the historical and future dates. Based on the provided thresholds, the magnitude of this warming is considered modest. The climate model suggests a noticeable but not extreme shift in local temperature under these conditions.** | **Q: For an influenza season in North Carolina, Missouri, Arizona, and Vermont beginning around October 7th, 2025, what is the comprehensive epidemiological forecast for hospital prevalence, assuming a high basic reproduction number (R0) of 2.2, moderate seasonality, and a low 10% population-wide prior immunity? Please analyze the outbreak's expected trajectory, considering its peak severity, and timing.**

**A: The forecast points to a severe and rapidly escalating influenza season. The outbreak is projected to begin with an extremely rapid growth phase. The peak magnitude of hospitalizations is expected to be very high, reaching a median of approximately 2234 cases. This significant peak is anticipated to arrive approximately 7.3 weeks after the season's start. Collectively, these indicators suggest a major and sustained wave of influenza.** |

Figure 3: Example questions and answers from our benchmark dataset for climate and epidemiology.

$(CO_2, CH_4, BC, SO_2)$ as inputs and 2-meter surface temperature as output, it captures anthropogenic warming drivers while targeting policy-relevant climate variables. This enables rapid assessment of emission pathways and climate risk evaluation. For epidemiology, we employ GLEAM-AI, a stochastic emulator capable of reproducing complex influenza transmission patterns in the United States (Zahedi et al., 2024; Wu et al., 2023). This emulator accurately replicates the mechanistic disease dynamics of the Global Epidemic and Mobility model (GLEAM) (Balcan et al., 2010; Chinazzi et al., 2024), a stochastic, age-stratified, metapopulation model integrating high-resolution population data, age-stratified social mixing dynamics, human mobility, and disease transmission. Our benchmark considers a Susceptible-Latent-Infectious-Removed-like compartmental model (Zahedi et al., 2024) simulating seasonal influenza outbreaks in the U.S., previously validated in influenza forecasting efforts (Mathis et al., 2024). We vary the following parameters in GLEAM-AI: basic reproduction number (R0), the strength of the seasonality, the level of initial residual immunity in the population, and the presumed starting date of the outbreak. For urban planning, we utilize the SUMO simulator (Lopez et al., 2018) to model traffic dynamics.

Using these emulators, we design a three-stage benchmark generation pipeline. First, the LLM receives simulator handbooks and identifies core functionalities, generating textual templates describing input-output relationships. These human-readable templates contain placeholders for input parameters and simulation results, stored in JSON format. Second, we programmatically sample input parameters from scientifically plausible ranges, execute simulators, and populate template placeholders with parameter-output pairs. Finally, we prompt the LLM to formulate open-ended questions requiring quantitative reasoning and qualitative interpretation, directly referencing numerical data. Crucially, the LLM generates ground-truth answers derived solely from simulation evidence, ensuring factual consistency and eliminating external knowledge or hallucinations. Figure 3 shows example questions and answers.

## 4 EXPERIMENTS

### 4.1 EXPERIMENTAL SETUP

We evaluate our proposed SimulRAG framework and UE+SBA method across two scientific domains: climate and epidemiological modeling. We employ GPT-4o and Claude-3.5 Sonnet as backbone LLM models. Our objective is demonstrating that SimulRAG improves answer informativeness and factuality compared to traditional RAG baselines, while UE+SBA enhances efficiency and quality for claim-level generation.

**Datasets.** We construct two benchmark datasets described in Section 3.4. The climate modeling dataset contains 200 free-form questions covering various climate phenomena. The epidemiological modeling dataset comprises 200 free-form questions concerning disease spread dynamics and plausible future scenarios. Detailed dataset examples and statistics are provided in Appendix A.3.

**Claim-level evaluation.** We assess generated answer quality at the atomic claim level. Each answer undergoes decomposition into constituent atomic claims via structured prompting. Claims are subsequently evaluated for correctness through a rigorous two-stage verification process. First, an LLM judge evaluates simulator-related claims using ground truth references. Subsequently, we

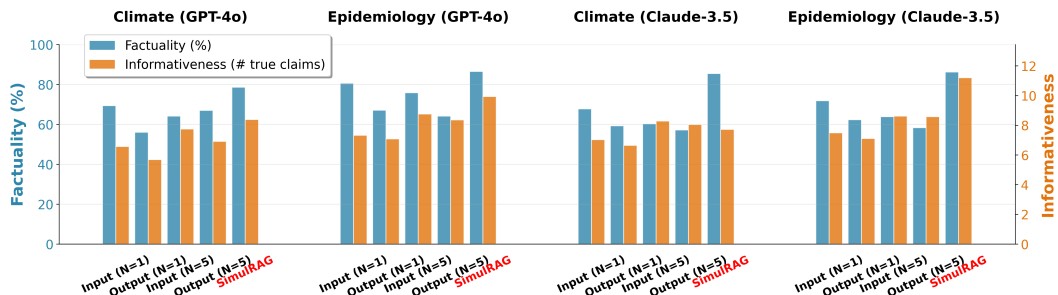

Figure 4: Factuality and informativeness comparison across RAG methods, with baselines where answers are sampled $N = 1$ and $N = 5$ times. SimulRAG ($N = 5$) consistently outperforms baseline methods on both evaluation metrics, showing superior capability for generating informative and factual long-form scientific answers.

collaborate with PhD students in the relevant fields to manually assess remaining claims based on correctness and relevance to the posed question. Claims receive true labels only when both correct and relevant to the question.

**Baselines.** We evaluate SimulRAG against established RAG baselines: (1) **Input-layer integration** (Ram et al., 2023) directly concatenates queries with retrieved context as LLM input; (2) **Output-layer integration** (Yu et al., 2023) refines generated answers using retrieval context post-processing. These baselines require adaptation for scientific simulators. We adapt them through our simulator retrieval interface to extract textual context from simulation outputs. This enables fair comparison, demonstrating SimulRAG's superior performance in answer informativeness and factual accuracy.

For claim-level generation, we evaluate UE+SBA against baselines: (1) **Random**: Randomly selects claims for simulator verification. (2) **Verbalized**: Uses LLM verbalized uncertainty estimates (Lin et al., 2022; Tian et al., 2023). (3) **Uncertainty**: Selects claims using confidence scores (Jiang et al., 2024). All methods use identical claim decomposition and answer regeneration procedures from Section 3.3. This ensures fair comparison and demonstrates UE+SBA's effectiveness in selecting valuable claims for verification and modification.

**Evaluation metrics.** We assess SimulRAG and baselines using informativeness and factuality metrics for answers. Informativeness counts unique true claims within answers. Factuality measures the proportion of true claims across all generated claims.

For claim-level methods, we evaluate efficiency and quality by varying verification budgets (percentage of claims selected for verification updates). Updated claims are verified against ground truth references. We compute F1, AUPR, and AUROC metrics for updated claim sets using uncertainty estimation scores. Higher values indicate superior claim verification and update quality. F1 uses balanced scoring to identify thresholds where precision approximates recall. Corresponding values appear in Appendix A.4.1.

## 4.2 SimulRAG Performance

We first assess the effectiveness of the SimulRAG framework for long-form scientific QA. Figure 4 demonstrates the informativeness and factuality of different RAG methods across two benchmarks: Climate and Epidemiology, using two LLM models: GPT-4o and Claude-3.5. SimulRAG consistently outperforms the existing baselines across both evaluation metrics. For informativeness, SimulRAG achieves 30.4% more unique true claims on average over the best baseline. For factuality, SimulRAG attains 16.3% higher proportions of true claims over the best baseline. These results substantiate SimulRAG's effectiveness in generating high-quality long-form scientific answers through systematic verification and updating of atomic claims using retrieved simulation outputs.

| Benchmark | Metric | Method | GPT-4o | | | Claude 3.5 | | |
|---|---|---|---|---|---|---|---|---|
| | | | 15% | 25% | 45% | 15% | 25% | 45% |
| Climate | F1 Score | Random | 0.6685 | 0.6794 | 0.7039 | 0.7323 | 0.7419 | 0.7511 |
| | | Verbalized | 0.6519 | 0.647 | 0.7393 | 0.6766 | 0.7251 | 0.723 |
| | | Uncertainty | 0.6894 | 0.7113 | 0.744 | 0.7384 | 0.7509 | 0.793 |
| | | UE+SBA | **0.702** | **0.731** | **0.7725** | **0.7511** | **0.7768** | **0.8242** |
| | AUPR | Random | 0.7144 | 0.7383 | 0.7594 | 0.7876 | 0.7932 | 0.8048 |
| | | Verbalized | 0.6745 | 0.7071 | 0.7461 | 0.7532 | 0.7731 | 0.8023 |
| | | Uncertainty | 0.7391 | 0.7522 | 0.7714 | 0.777 | 0.7817 | 0.8092 |
| | | UE+SBA | **0.746** | **0.7629** | **0.7871** | **0.7879** | **0.8039** | **0.8297** |
| | AUROC | Random | 0.6013 | 0.6267 | 0.6618 | 0.6672 | 0.6793 | 0.7031 |
| | | Verbalized | 0.5522 | 0.6006 | 0.6659 | 0.6172 | 0.66 | 0.7277 |
| | | Uncertainty | 0.6405 | 0.6645 | 0.7040 | 0.6818 | 0.6975 | 0.7446 |
| | | UE+SBA | **0.6531** | **0.6854** | **0.7253** | **0.7002** | **0.729** | **0.7709** |
| Epidemiology | F1 Score | Random | 0.5898 | 0.6157 | 0.6581 | 0.6961 | 0.7042 | 0.7374 |
| | | Verbalized | 0.5654 | 0.627 | 0.6953 | 0.6804 | 0.7421 | 0.731 |
| | | Uncertainty | 0.6103 | 0.644 | 0.7043 | 0.706 | 0.7319 | 0.7848 |
| | | UE+SBA | **0.6431** | **0.6957** | **0.8155** | **0.7231** | **0.7594** | **0.8207** |
| | AUPR | Random | 0.6727 | 0.7053 | 0.7555 | 0.7428 | 0.7687 | 0.8097 |
| | | Verbalized | 0.5972 | 0.6534 | 0.7402 | 0.7221 | 0.7599 | 0.8206 |
| | | Uncertainty | 0.6833 | 0.7243 | 0.7799 | 0.7482 | 0.7798 | 0.826 |
| | | UE+SBA | **0.7239** | **0.7754** | **0.8424** | **0.7739** | **0.8113** | **0.859** |
| | AUROC | Random | 0.6454 | 0.6753 | 0.7295 | 0.5716 | 0.6045 | 0.6723 |
| | | Verbalized | 0.6295 | 0.6834 | 0.7773 | 0.5358 | 0.5993 | 0.7175 |
| | | Uncertainty | 0.6789 | 0.7325 | 0.8005 | 0.5972 | 0.6489 | 0.7397 |
| | | UE+SBA | **0.7308** | **0.7906** | **0.8634** | **0.6338** | **0.6971** | **0.7916** |
| Urban | F1 Score | Random | 0.7919 | 0.7980 | 0.8105 | 0.6801 | 0.7030 | 0.7525 |
| | | Verbalized | 0.7777 | 0.7859 | 0.8022 | 0.6863 | 0.7105 | 0.7545 |
| | | Uncertainty | 0.7949 | 0.7993 | 0.8151 | 0.6839 | 0.7082 | 0.7538 |
| | | UE+SBA | **0.8078** | **0.8224** | **0.8290** | **0.7173** | **0.7613** | **0.8222** |
| | AUPR | Random | 0.8328 | 0.8359 | 0.8548 | 0.7575 | 0.7919 | 0.8467 |
| | | Verbalized | 0.7880 | 0.8021 | 0.8265 | 0.7365 | 0.7733 | 0.8329 |
| | | Uncertainty | 0.8292 | 0.8338 | 0.8529 | 0.7604 | 0.7921 | 0.8445 |
| | | UE+SBA | **0.8444** | **0.8569** | **0.8715** | **0.7987** | **0.8469** | **0.9019** |
| | AUROC | Random | 0.6000 | 0.6029 | 0.6280 | 0.5892 | 0.6146 | 0.6675 |
| | | Verbalized | 0.5157 | 0.5304 | 0.5541 | 0.5817 | 0.6109 | 0.6657 |
| | | Uncertainty | 0.5982 | 0.6005 | 0.6223 | 0.5914 | 0.6178 | 0.6721 |
| | | UE+SBA | **0.6346** | **0.6537** | **0.6774** | **0.6475** | **0.7091** | **0.7833** |

Table 1: Performance comparison of claim-level generation methods across three benchmarks and two LLM models. Results show F1 Score, AUPR, and AUROC for 3 different verification budgets (15%, 25%, 45%). The 15% budget means 15% of claims will be verified and possibly updated. Best performing methods are bolded.

## 4.3 UE+SBA PERFORMANCE

We evaluate the efficiency and quality of uncertainty estimation scores and simulator boundary assessment (UE+SBA) for claim-level generation. Table 1 presents the performance comparison of different claim-level generation methods across climate and epidemiology benchmarks, using GPT-4o and Claude-3.5 models. We report F1 score, AUPR, and AUROC across 3 different verification budgets (15%, 25%, 45%). For example, a 15% budget indicates verifying and updating 15% of claims selected by each method. It can be observed that UE+SBA consistently outperforms all baseline methods across all metrics, models, benchmarks, and budgets. For AUPR, UE+SBA achieves up to 6.2% absolute improvements over the best baseline. For AUROC, UE+SBA attains up to 6.3% absolute improvements over the best baseline. Meanwhile, the Uncertainty baseline consistently ranks second, demonstrating the effectiveness of uncertainty estimation for claim selection, a key component of UE+SBA. The next section analyzes simulator boundary assessment (SBA) effectiveness in detail.

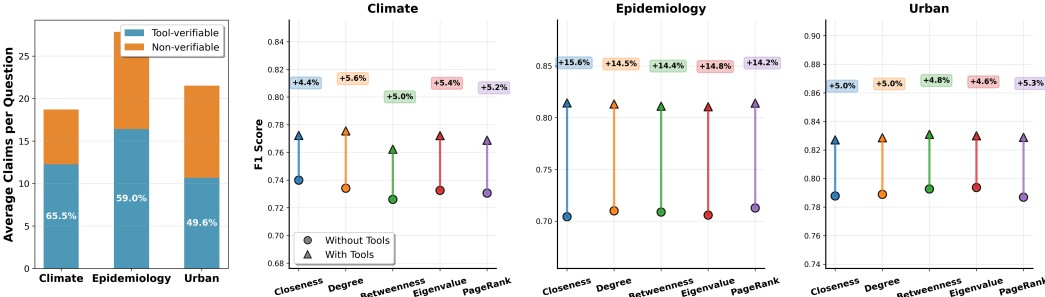

Figure 5: Left: Proportion of SBA-selected claims verifiable by simulator. Right: Performance comparison between Uncertainty and UE+SBA across five uncertainty estimation scores on climate science, epidemiology, and urban planning benchmarks.

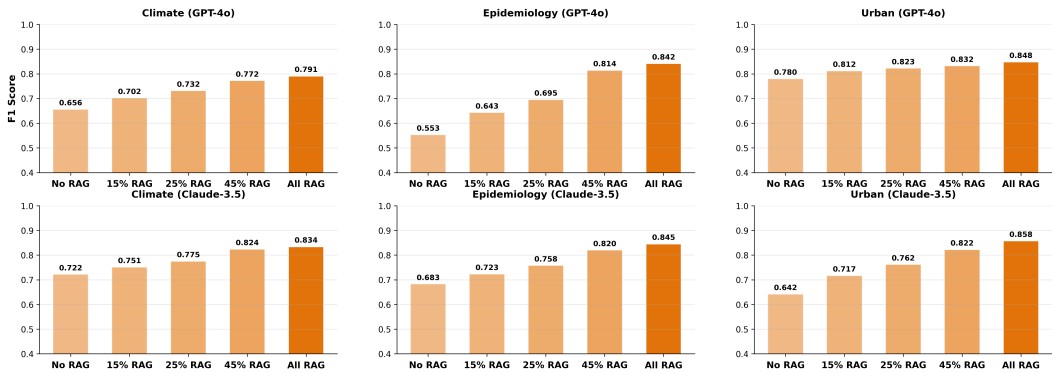

Figure 6: F1 score comparison of SimulRAG across RAG verification budgets (15%, 25%, 45%) versus no-RAG and all-RAG on three benchmarks.

## 4.4 SIMULATOR BOUNDARY ASSESSMENT (SBA) EFFECTIVENESS

To evaluate SBA effectiveness, we visualize the proportion of SBA-selected claims verifiable by the simulator. Figure 5 left demonstrates that SBA effectively filters out non-verifiable claims. For climate, SBA filters out 34.5% of claims on average. For epidemiology, SBA filters out 41% on average. Human evaluation confirms most filtered claims are indeed non-verifiable by the simulator. Thus, SBA reduces unnecessary claim-level verification and updates. Additionally, we compare UE+SBA with Uncertainty across 5 uncertainty estimation scores (Jiang et al., 2024): (1) Closeness, (2) Degree, (3) Betweenness, (4) Eigenvalue, (5) PageRank. Figure 5 right presents the performance comparison between Uncertainty and UE+SBA across these uncertainty estimation scores. Results show UE+SBA consistently outperforms Uncertainty alone across all scores on climate science and epidemiology benchmarks. This demonstrates SBA effectively complements uncertainty estimation to select more valuable claims for verification, yielding higher quality updated claim sets.

## 4.5 ABLATION STUDIES

We compare SimulRAG performance across RAG verification budgets (15%, 25%, 45%) against no-RAG and all-RAG in Figure 6 using F1 score. Results show SimulRAG with 15% RAG significantly outperforms no-RAG across both benchmarks and models. SimulRAG with 45% RAG achieves performance comparable to all-RAG scenarios. This demonstrates SimulRAG effectively balances efficiency and quality by selectively verifying a subset of valuable claims rather than all. This targeted approach maximizes verification impact while minimizing computational overhead.

## 5 CONCLUSION

In this work, we introduce SimulRAG, a simulator-based retrieval-augmented generation (RAG) framework for long-form scientific question answering. SimulRAG proposes a generalized simulator retrieval interface to transform between textual and numerical modalities, enabling seamless integration of scientific simulators into RAG systems. To improve answer generation quality, we present a claim-level generation method that decomposes long-form answers into atomic claims for fine-grained verification and updates. To efficiently verify and update claims, we utilize uncertainty estimation scores and simulator boundary assessment (UE+SBA) to selectively identify valuable claims for verification. Finally, we construct a long-form scientific QA benchmark covering climate science and epidemiology. Extensive experiments verify the effectiveness of our proposed Simul-RAG framework and UE+SBA method. One limitation of SimulRAG is assuming all questions relate to available simulators. Future work could develop automatic question-simulator relevance detection to avoid failed retrieval when prompts are unrelated and enable broader applicability across diverse scientific domains.

## Reproducibility Statement

We will release all code, simulators, and benchmark dataset for reproducibility. The code repository is submitted as supplementary material and will be publicly available upon acceptance. Implementation details are provided in the repository readme file.

## Ethics Statement

Despite two-stage verification and updating, SimulRAG may still generate factually incorrect answers. Fully trusting this automated system without human expert review could be risky for public health and climate policy decision making. We recommend human expert review as a necessary safeguard.

## LLM Usage

We only use LLMs to polish writing, not for content or idea generation.

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

# A  APPENDIX

## A.1  SIMULRAG ALGORITHM

The SimulRAG algorithm is shown in Algorithm 1. The key functions used in the algorithm are defined as follows:

- $I(S, q, h)$: Simulator retrieval interface that extracts parameters from question $q$ using handbook $h$, executes simulator $S$, and converts outputs to textual context $d$.
- LLM$(q)$: Generates diverse initial answer given question $q$.
- Decompose$(a_j)$: Decomposes answer $a_j$ into atomic claims following factual statement extraction principles.
- Merge$(\cdot)$: Merges and deduplicates claim sets from multiple answers using semantic equivalence detection.
- UE$(c_i, \mathcal{A}, \mathcal{C})$: Uncertainty estimation that computes confidence score for claim $c_i$ using entailment graph centrality.
- bound$(c_i, h)$: Boundary assessment function that determines whether claim $c_i$ is verifiable by simulator using handbook $h$.
- VerifyAndUpdate$(c_i, d)$: Verifies claim $c_i$ against simulation context $d$ and updates if contradictory or aligns if consistent.
- GenerateCoherentAnswer$(q, \mathcal{C}_{\text{final}})$: Synthesizes final answer from high-confidence claims in response to question $q$.

## A.2  PROMPT AND TEMPLATE DETAILS

Here we provide detailed prompts and templates used in our SimulRAG framework.

### A.2.1  CLAIM DECOMPOSITION PROMPT

After the answer set is recieved, the Claim Decomposition Prompt is used to decompose each answer into multiple claims. It preserves the semantic meaning of the original answer and focuses on single

---

**Algorithm 1** SimulRAG Framework

1: **Input:** Question $q$, Scientific simulator $S$, Handbook $h$, Thresholds $\tau, \kappa$
2: **Output:** Refined answer $a'$
3: $d \leftarrow I(S, q, h)$
4: $\mathcal{A} \leftarrow \{a_1, \ldots, a_m\}$ where $a_j \leftarrow$ LLM$(q)$
5: **for** $j = 1$ to $m$ **do**
6: $\quad \{c_{j1}, \ldots, c_{jn_j}\} \leftarrow$ Decompose$(a_j)$
7: **end for**
8: $\mathcal{C} \leftarrow$ Merge$(\bigcup_j \{c_{j1}, \ldots, c_{jn_j}\})$
9: $\mathcal{C}_{\text{selected}} \leftarrow \emptyset$
10: **for** $c_i \in \mathcal{C}$ **do**
11: $\quad$ conf$(c_i) \leftarrow$ UE$(c_i, \mathcal{A}, \mathcal{C})$
12: $\quad$ **if** conf$(c_i) < \tau$ **and** bound$(c_i, h) = 1$ **then**
13: $\quad\quad \mathcal{C}_{\text{selected}} \leftarrow \mathcal{C}_{\text{selected}} \cup \{c_i\}$
14: $\quad$ **end if**
15: **end for**
16: **for** $c_i \in \mathcal{C}_{\text{selected}}$ **do**
17: $\quad c_i \leftarrow$ VerifyAndUpdate$(c_i, d)$
18: $\quad$ conf$(c_i) \leftarrow 1$
19: **end for**
20: $\mathcal{C}_{\text{final}} \leftarrow \{c_i \in \mathcal{C} \mid$ conf$(c_i) \geq \kappa\}$
21: $a' \leftarrow$ GenerateCoherentAnswer$(q, \mathcal{C}_{\text{final}})$
22: **return** $a'$

---

setting related to the answer, not the repetition of the question. We adapt the prompt from Jiang et al. (2024), the prompt is given as follows:

> Please deconstruct the following paragraph into the smallest possible standalone self-contained facts without semantic repetition, and return the output as a jsonl, where each line is {claim:[CLAIM]}.
>
> CRITICAL: Extract ONLY the 8-12 MOST IMPORTANT claims. Be extremely selective. Focus ONLY on:
>
> - Direct answers to the specific question asked
>
> - Specific numerical values, percentages%, or measurements
>
> - Key causal relationships (A causes B)
>
> - Critical scientific conclusions
>
> STRICTLY AVOID:
>
> - General background information
>
> - Basic definitions (what SO2 is, what SSP scenarios are, etc.)
>
> - Procedural explanations
>
> - Location descriptions
>
> - Any claim that doesn't directly address the question
>
> Each claim must be essential to answering the question. If unsure whether to include a claim, DON'T include it.
>
> The input is: {original_text}
>
> Response:

### A.2.2 CLAIM MERGING PROMPT

Each answer is decomposed into a claim set. The Claim merging Prompt is used merge all claim set into one single claim set. It is ensured all deplicated claims are merged and all claims are independent with each other. We adopt the idea from Jiang et al. (2024) and designed the prompt given as follows:

> You are given two sets of claims. Find which claims in Set B are already covered by claims in Set A.
>
> Set A (Existing claims): {existing_claim_set}
>
> Set B (New claims): {new_claim_set}
>
> For each claim in Set B, check if it says essentially the same thing as any claim in Set A (i.e. semantic equivalence even if worded differently). Should be equivalent in meaning, if A said something turns up while B said something turns down, they are not equivalent. If A said something turns up and B said the same thing goes up, they are equivalent.
>
> You must respond with ONLY a valid Json array of pairs. Each pair is [existing_index, new_index] where Set A[existing_index] covers Set B[new_index].
>
> Examples:
>
> - If Set A[0] covers Set B[1] and Set A[2] covers Set B[0]: [[0, 1], [2, 0]]
>
> - If no claims match: []

```
- If Set A[1] covers Set B[0]: [[1, 0]]

Response:
```

### A.2.3  SCIENTIFIC BOUNDARY ANALYSIS PROMPT

As mentioned in 3.3, given the scientific simulator's handbook and the claim. The Scientific Boundary Analysis Promptis used to identify if the claim could be verified or updated by the existing tools. The prompt is given as follows:

```
You are an expert in [Dataset] science and computational tools.  You
will evaluate whether [Dataset]-related claims can be verified using
available [Dataset] simulation tools.

**AVAILABLE CLIMATE TOOLS:** {tools_handbook}

**EVALUATION TASK:** For each claim provided, you need to determine how
well the available tools can help verify or assess the accuracy of that
claim.

**SCORING CRITERIA:**

- **0**:  The claim cannot be verified or assessed using any of the
available tools

- Examples:  Claims about non-climate topics, general policy
statements, claims requiring data/tools not available, claims that
touch on climate aspects but cannot be directly verified with the
specific tools provided

- **1**:  The claim can be directly and comprehensively verified or
assessed using the available tools

- Examples:  Claims about temperature changes, climate scenarios,
aerosol/greenhouse gas impacts, geographic classifications, specific
quantitative climate predictions

**RESPONSE FORMAT:** Respond with ONLY a JSON object containing a
single key "tool_confidence" with a value of 0 or 1.

Example:  {"tool_confidence": 1}

**YOUR TASK:**

Question:  {question} Claim:  {claim}

Evaluate how well the available climate simulation tools can verify or
assess this specific claim in the context of the given question.

Response:
```

### A.2.4  VERIFICATION AND UPDATING PROMPT

The Verification and Updating Prompt is used to ask model use simulation's textual content to update and verify each selected claim. It will identifiy the claim is included in the content or not, and perform update when necessary. The prompt is given as follows:

```
You are a fact-checking assistant.  You have been given a claim and
some quantitative context information.  Your task is to analyze the
relationship between the claim and the RAG context to see if you should
update the claim or not.  You should assume the RAG context is 100%
correct and accurate.

INSTRUCTIONS:
```

1.  First, determine if the RAG context contains information relevant
to the claim's topic (set "is_included")

2.  If relevant, check if the claim should be updated for better
accuracy (set "should_update")

3.  If updating, modify the claim directly - do not generate new claims
or unrelated content.

4.  Only update the related part of the claim, don't add extra
information (i.e.  if the claim is related to A, and the rag provides
information about all of A, B, C, then you should update the parts
related to A only, but not B or C)

5.  Keep updates as minimal as possible and focused on improving
accuracy

6.  You may need to do calculations from the RAG context to perform the
update, it is required to do and please carefully do the numerical
calculations.

7.  Skip the update and mark it as not included if the claim is not
related to the RAG context.

DECISION CRITERIA:

- "is_included":  true if RAG context discusses the same
topic/concept/domain as the claim, false if completely unrelated

- "should_update":  true only if the claim has incorrect/incomplete
information that RAG context can improve

Respond in JSON format:  {{ "is_included":  true/false, "should_update":
true/false, "updated_claim":  "the updated claim text (only if
should_update is true)" }}

YOUR TASK INPUT: CLAIM TO EVALUATE: {claim}

RAG CONTEXT: {textual_context}

Response:

### A.2.5   FINAL ANSWER PROMPT

The Final Answer Prompt is used to ask model use selected claims to generate the final answer. The prompt is given as follows:

You are an expert answering a complex question based on provided
factual claims.

QUESTION: {question}

AVAILABLE CLAIMS: {claims_text}

TASK: Generate a comprehensive and accurate answer to the question
using only the information provided in the claims above.

REQUIREMENTS:

- Use only the factual information from the provided claims

- Synthesize the claims into a coherent, well-structured answer

- If claims conflict, prioritize the most specific and detailed
information

- If the claims don't fully address the question, acknowledge the
limitations

```
    - Do not add information beyond what's provided in the claims

    Generate a clear, comprehensive answer:
```

### A.2.6 TEXTUAL CONTEXT TEMPLATE

As described in 3.2, we use textual context template to translate the simulation result into human-readable context. It will fill in the placeholder fields with simulation results to complete the text. An example of climate textual context template is given as follows:

```
"Query":  "If CO2 emissions increase by {{delta_CO2}}% and CH4 emissions
increase by {{delta_CH4}}% in {{year}} under the {{setting}} scenario,
what would be the average temperature for {{city_name}}?  Also, is
{{city_name}} located on land or sea?"

"Result":  "With a {{delta_CO2}}% increase in CO2 and {{delta_CH4}}%
increase in CH4, the average temperature for {{city_name}} in {{year}}
under the {{setting}} scenario would be {{greenhouse_temp}}°C. This
location is on {{land_sea_result}}."
```

And an example of epidemiology textual context template is given as follows:

```
"Query":  "What is the projected epidemiological landscape for
{{target_metric}} in {{target_states}} for an influenza season
initiating around {{starting_date}}, assuming a basic reproduction
number (R0) of {{r0_value}}, a {{seasonality_level}} influence, and an
initial population immunity level of {{prior_immunity_level}}?"

"Result":  "Projected Epidemiological Landscape for
{{target_metric}}:{{simulation_outlook}}"
```

### A.3 DATASET STATISTICS AND EXAMPLES

We include examples of questions, ground truth answers, and the corresponding groundtruth claim sets generated on the two benchmark datasets used in our experiments. We also provide statistics of the two datasets.

### A.3.1 EXAMPLE CLIMATE QUESTIONS

Below gives example questions generated in our climate benchmark:

---

**Question**: For the city of Chaiwu, what is the total temperature change projected between its historical average in 1994 and a future scenario in 2041 under ssp585 conditions, where CO2 emissions increase by 47.31% and CH4 emissions increase by 36.46%? Does this change represent a warming or cooling trend, and would its magnitude be considered modest (less than 1.0°C) or significant (1.0°C or greater)?

**Reference Answer**: The total projected temperature change is an increase of 1.08°C, which indicates a clear warming trend for Chaiwu between the two dates. Based on the provided thresholds, the magnitude of this change is considered significant. This suggests a notable shift in the local climate under the specified emissions scenario.

**Reference Answer Claims**:

- The total projected temperature change is an increase of 1.08°C.
- There is a clear warming trend for Chaiwu between the two dates.
- The magnitude of the temperature change is considered significant based on the provided thresholds.

---

- There is a notable shift in the local climate under the specified emissions scenario.

**Question**: For Suresnes in 2050, under the ssp245 scenario with CO2 emissions increasing by 23.83% and CH4 by 39.2%, what is the projected temperature? Based on this projection, would the local climate be considered cold (below 5°C), mild (5°C to 15°C), or warm (above 15°C)? Also, confirm if the location is terrestrial.

**Reference Answer**: The projected temperature for Suresnes is 7.9°C, which is classified as mild according to the given temperature ranges. The location is confirmed to be terrestrial. The projected climate does not fall into the cold or warm categories.

**Reference Answer Claims**:

- The projected temperature for Suresnes is 7.9°C.
- The projected temperature of 7.9°C for Suresnes is classified as mild.
- The projected climate for Suresnes does not fall into the cold category.
- The projected climate for Suresnes does not fall into the warm category.

**Question**: For the city of Baldwin in 2072 under the ssp245 scenario, compare two distinct emission modification strategies: one where CO2 emissions increase by 13.4% and CH4 emissions increase by 6.09%, and another where SO2 emissions decrease by 13.28% and Black Carbon emissions are modified by -11.75% at [(-73.6075, 40.6511)]. Which strategy results in a warmer local climate? What is the temperature difference between the two scenarios? Is this difference considered negligible (less than 0.2°C), modest (0.2°C to 1.0°C), or significant (over 1.0°C)?

**Reference Answer**: The scenario involving an increase in greenhouse gas emissions results in a warmer local climate compared to the aerosol modification scenario. The temperature difference between the two strategies is 0.26°C, which is considered a modest divergence based on the given thresholds. This indicates that the specified greenhouse gas increases have a more pronounced warming impact than the aerosol changes in this particular forecast.

**Reference Answer Claims**:

- An increase in greenhouse gas emissions results in a warmer local climate compared to aerosol modification.
- The temperature difference between the greenhouse gas emissions scenario and the aerosol modification scenario is 0.26°C.
- The 0.26°C temperature difference is considered a modest divergence based on the given thresholds.
- Greenhouse gas increases have a more pronounced warming impact than aerosol changes in this forecast.

**Question**: For Chongzuo, compare the historical average temperature in 1992 with the projection for 2083 under the ssp245 scenario, assuming a 29.37% increase in CO2 and a 48.75% increase in CH4 emissions. What is the total temperature change? Based on this change, does this represent a minor (less than 1°C), significant (1°C to 2°C), or severe (over 2°C) warming trend?

**Reference Answer**: The total projected temperature change is an increase of 1.18°C, which indicates a distinct warming trend for Chongzuo. Based on the provided thresholds, this level

of temperature rise is classified as significant. This suggests a considerable shift in the local climate between the historical and future periods under this emissions scenario.

**Reference Answer Claims**:

• The total projected temperature change is an increase of 1.18°C for Chongzuo.

• The 1.18°C temperature increase indicates a distinct warming trend for Chongzuo.

• This level of temperature rise is classified as significant based on the provided thresholds.

• The temperature increase suggests a considerable shift in the local climate between historical and future periods under this emissions scenario.

---

**Question**: For Correia Pinto in the year 2072 under the ssp585 scenario, compare two potential interventions: an aerosol modification with a 6.24% change in SO2 and -19.46% change in BC at points [(-50.4, -27.5833)], versus a greenhouse gas intervention with a 44.57% change in CO2 and 8.5% change in CH4. Which intervention leads to a warmer local climate? What is the temperature difference between them? Would this difference be considered negligible (less than 0.2°C), modest (0.2°C to 1.0°C), or significant (over 1.0°C)?

**Reference Answer**: The greenhouse gas intervention results in a warmer local climate compared to the aerosol modification. The temperature difference between the two scenarios is 0.61°C, which is considered a modest impact based on the provided thresholds. This indicates that the choice between these two strategies has a noticeable effect on the projected local temperature.

**Reference Answer Claims**:

• The greenhouse gas intervention results in a warmer local climate compared to the aerosol modification.

• The temperature difference between the greenhouse gas intervention and aerosol modification scenarios is 0.61°C.

• The temperature difference of 0.61°C is considered a modest impact based on the provided thresholds.

• The choice between greenhouse gas intervention and aerosol modification strategies has a noticeable effect on the projected local temperature.

A.3.2  EXAMPLE EPIDEMIOLOGY QUESTIONS

Below gives example questions generated in our epidemiology benchmark:

---

**Question**: For an upcoming influenza season in North Carolina and Massachusetts commencing around October 4th, 2022, what is the comprehensive epidemiological forecast for hospital prevalence, assuming a high R0 of 2.6, moderate seasonality, and a low 10% prior immunity? Please assess the expected trajectory, including the outbreak's peak severity, and timing.

**Reference Answer**: The forecast indicates an extremely severe and rapidly evolving outbreak. The season is projected to begin with an explosive growth phase. The peak magnitude of hospitalizations is expected to be exceptionally high, reaching a median value of around 11,941. This severe peak is anticipated to materialize very quickly, arriving just 5.4 weeks after the season's onset. This trajectory points to a fast-moving, high-burden wave that will develop with remarkable speed.

**Reference Answer Claims**:

- The forecast indicates an extremely severe and rapidly evolving outbreak.
- The season is projected to begin with an explosive growth phase.
- The peak magnitude of hospitalizations is expected to reach a median value of around 11,941.
- The severe peak is anticipated to materialize 5.4 weeks after the season's onset.
- The trajectory points to a fast-moving, high-burden wave.

**Question**: For an influenza season in North Carolina, Missouri, Arizona, and Vermont beginning around October 7th, 2022, what is the comprehensive epidemiological forecast for hospital prevalence, assuming a high basic reproduction number (R0) of 2.2, moderate seasonality, and a low 10% population-wide prior immunity? Please analyze the outbreak's expected trajectory, considering its peak severity, and timing.

**Reference Answer**: The forecast points to a severe and rapidly escalating influenza season. The outbreak is projected to begin with an extremely rapid growth phase. The peak magnitude of hospitalizations is expected to be very high, reaching a median of approximately 2234 cases. This significant peak is anticipated to arrive approximately 7.3 weeks after the season's start. Collectively, these indicators suggest a major and sustained wave of influenza.

**Reference Answer Claims**:

- The influenza season is forecasted to be severe and rapidly escalating.
- The peak magnitude of hospitalizations is expected to reach a median of approximately 2234 cases.
- The peak of hospitalizations is anticipated to arrive approximately 7.3 weeks after the season's start.
- Indicators suggest a major and sustained wave of influenza.

**Question**: For an influenza season in Colorado, North Carolina, Vermont, and South Carolina beginning around September 23, 2022, what is the comprehensive epidemiological forecast for hospital prevalence, assuming a high basic reproduction number (R0) of 2.6, no seasonality, and a low population-wide prior immunity of 10%? Please analyze the projected trajectory, considering the outbreak's peak severity, and timing.

**Reference Answer**: The forecast points to an extremely severe and explosive outbreak scenario. The season is projected to begin with an unprecedentedly rapid growth phase. The peak magnitude of hospitalizations is expected to be exceptionally high, reaching a median value of approximately 4,308 concurrent cases. This severe peak is forecast to arrive very quickly, materializing just 4.1 weeks after the season's onset. Taken together, these indicators suggest a rapid, overwhelming wave of infections that escalates to a severe peak in just over a month.

**Reference Answer Claims**:

- The peak magnitude of hospitalizations is expected to reach a median value of approximately 4,308 concurrent cases.
- The severe peak of hospitalizations is forecast to materialize 4.1 weeks after the season's onset.
- The outbreak is expected to begin with an unprecedentedly rapid growth phase.
- The indicators suggest a rapid, overwhelming wave of infections.

**Question**: What is the comprehensive epidemiological forecast for hospital prevalence in Iowa, Florida, and Michigan, for an influenza season starting around September 22nd, 2022? This scenario assumes a high R0 of 2.2, moderate seasonality, and a low 10% prior immunity in the population. Please consider severity of the outbreak, and its timing.

**Reference Answer**: The epidemiological forecast indicates an exceptionally severe and rapidly developing influenza season. The outbreak is expected to begin with an explosive growth phase. This rapid escalation is projected to culminate in a very high peak of hospitalizations, with a median magnitude of approximately 3115 concurrent cases. The peak of this intense wave is forecast to arrive relatively quickly, about 7.1 weeks after the season's onset. This trajectory suggests a severe, front-loaded epidemic wave with a significant and swift impact.

**Reference Answer Claims**:

- The influenza season is expected to be exceptionally severe and rapidly developing.
- The outbreak is expected to begin with an explosive growth phase.
- The peak of hospitalizations is projected to have a median magnitude of approximately 3115 concurrent cases.
- The peak of the influenza wave is forecast to arrive about 7.1 weeks after the season's onset.
- The epidemic wave is expected to be severe and front-loaded with a significant and swift impact.

---

**Question**: For an upcoming influenza season in South Dakota, Massachusetts, Illinois, and Wyoming, what is the comprehensive epidemiological forecast for hospital prevalence given a start date of September 26, 2022, a high R0 of 2.4, strong seasonality, and an initial population immunity level of 20%? Please assess the expected trajectory, considering the peak's severity, and its timing.

**Reference Answer**: The epidemiological forecast for this scenario indicates a severe and sustained influenza wave. The season is expected to begin with a rapid growth phase. The peak magnitude of hospitalizations is projected to be very high, reaching a median value of approximately 999 cases. This severe peak is anticipated to materialize late in the season, occurring about 11 weeks after the start date. Overall, this trajectory points to a challenging season characterized by a rapid initial spread and a high, delayed peak.

**Reference Answer Claims**:

- The epidemiological forecast indicates a severe and sustained influenza wave.
- The influenza season is expected to begin with a rapid growth phase.
- The peak magnitude of hospitalizations is projected to reach a median value of approximately 999 cases.
- The severe peak in hospitalizations is anticipated to occur about 11 weeks after the start date.
- The influenza season is characterized by a rapid initial spread and a high, delayed peak.

### A.3.3 EXAMPLE URBAN PLANNING QUESTIONS

Below gives example questions generated in our urban planning benchmark:

**Question**: To combat downtown gridlock, the city is proposing a two-part strategy: 1) A 'Low-Speed Zone' that reduces speed limits by 27% to discourage through-traffic, and 2) Upgrading

the main corridor's traffic signal to a 'green wave' priority junction. What is the percentage change in average vehicle idling time, and is this reduction considered marginal (under 25%), moderate (25-75%), or profound (over 75%)? What is the percentage change in total $CO_2$ emissions, and is this reduction considered negligible (under 10%), significant (10-40%), or transformative (over 40%)? Based on these projected outcomes, how effectively does this strategy address both traffic flow efficiency and environmental sustainability goals?

**Reference Answer**: The combined strategy delivers exceptional improvements in both traffic efficiency and environmental quality. Average vehicle idling time is projected to decrease by 93.1%, which is a profound reduction indicating significantly smoother traffic flow and reduced congestion. Total $CO_2$ emissions are also expected to decrease by 33.99%, representing a significant positive impact on air quality. This intervention successfully achieves substantial co-benefits, effectively combating gridlock and advancing environmental sustainability goals by drastically reducing wasted fuel and emissions.

**Reference Answer Claims**:

- Average vehicle idling time is projected to decrease by 93.1%.
- Total $CO_2$ emissions are expected to decrease by 33.99%.
- The strategy delivers improvements in traffic efficiency.
- The strategy delivers improvements in environmental quality.
- The intervention reduces congestion.
- The intervention reduces wasted fuel.
- The intervention reduces emissions.
- The intervention advances environmental sustainability goals.

**Question**: For our 'Livable Downtown' initiative, a proposed 'road diet' on the main arterial involves reducing the road from 2 lanes to 1 to create a protected bike lane, and simultaneously lowering the speed limit by 26%. What is the percentage change in average vehicle travel time? Is this change considered minor (under 5% increase), moderate (5-15% increase), or significant (over 15% increase)? What is the percentage change in total $CO_2$ emissions, and is this reduction considered marginal (under 5%), notable (5-15%), or substantial (over 15%)? Considering both impacts, does this 'road diet' represent an acceptable trade-off for urban planners aiming for a 'Livable Downtown'?

**Reference Answer**: The proposed 'road diet' results in a significant increase in average vehicle travel time by 15.68%, indicating a clear impact on traffic flow for motorists. However, the initiative simultaneously delivers a notable environmental benefit, achieving a 14.91% reduction in total $CO_2$ emissions. This intervention presents a strategic trade-off, prioritizing sustainable transportation modes and air quality improvements over direct vehicular throughput, which is generally acceptable for achieving the broader goals of a 'Livable Downtown' by fostering a more pedestrian and bike-friendly environment.

**Reference Answer Claims**:

- The proposed 'road diet' results in a 15.68% increase in average vehicle travel time.
- The 'road diet' achieves a 14.91% reduction in total $CO_2$ emissions.
- The 'road diet' prioritizes sustainable transportation modes over direct vehicular throughput.
- The 'road diet' aims to improve air quality.
- The 'road diet' supports the broader goals of a 'Livable Downtown'.
- The 'road diet' fosters a more pedestrian and bike-friendly environment.

**Question**: A major downtown concert is expected to increase evening traffic demand by 22%. To mitigate the anticipated gridlock, the traffic authority is proposing to activate a 'green wave' priority junction along the main arterial leading to the venue. What is the net percentage change in average travel time? What is the net percentage change in vehicle idling, and would this be considered a minor (under 25%), significant (25-75%), or transformative (over 75%) impact? Based on these projected changes, does the 'green wave' strategy effectively counteract the concert's traffic surge and enhance overall urban mobility?

**Reference Answer**: The implementation of the 'green wave' priority junction demonstrates exceptional effectiveness in managing the increased traffic demand. Average travel time is projected to decrease by 27.08%, indicating that the intervention not only offsets the concert's impact but actively improves traffic flow beyond normal conditions. Furthermore, vehicle idling time is expected to decrease by 93.1%, representing a transformative reduction that will significantly reduce emissions and improve the driver experience. This strategy successfully counters the anticipated gridlock, ensuring a notably more efficient and sustainable urban mobility experience for the evening.

**Reference Answer Claims**:

- Average travel time is projected to decrease by 27.08% due to the 'green wave' priority junction.
- Vehicle idling time is expected to decrease by 93.1% with the implementation of the 'green wave' priority junction.
- The 'green wave' priority junction intervention offsets the concert's impact on traffic.
- The 'green wave' priority junction actively improves traffic flow beyond normal conditions.
- The reduction in vehicle idling time will significantly reduce emissions.
- The 'green wave' priority junction improves the driver experience by reducing idling time.
- The strategy counters anticipated gridlock effectively.
- The 'green wave' priority junction ensures a more efficient urban mobility experience for the evening.

---

**Question**: To create a 'Living Street' on the main commercial avenue, the city plans to reduce the speed limit by 32% to enhance pedestrian safety. To prevent this from causing major gridlock, they will also upgrade the central traffic light to a priority 'green wave' junction. What is the percentage change in average vehicle idling time, and is this reduction considered minor (under 25%), significant (25-75%), or drastic (over 75%)? What is the percentage change in total CO2 emissions, and is this reduction considered minor (under 10%), moderate (10-30%), or substantial (over 30%)? Does this combined intervention effectively achieve its goals for both traffic efficiency and environmental sustainability?

**Reference Answer**: The combined intervention demonstrates exceptional success in optimizing urban mobility and environmental quality. Average vehicle idling time sees a drastic reduction of 93.1%, indicating a highly efficient traffic flow despite the lower speed limit. Furthermore, total CO2 emissions decrease substantially by 35.25%. These results highlight that the 'green wave' priority junction effectively mitigates congestion from the speed reduction, leading to significant co-benefits for traffic efficiency, pedestrian safety, and air quality.

**Reference Answer Claims**:

- Average vehicle idling time sees a drastic reduction of 93.1%.
- Total CO2 emissions decrease substantially by 35.25%.

- The 'green wave' priority junction effectively mitigates congestion from the speed reduction.
- The intervention leads to significant co-benefits for traffic efficiency.
- The intervention leads to significant co-benefits for pedestrian safety.
- The intervention leads to significant co-benefits for air quality.

**Question**: The city is piloting a 'Smart Corridor' project on its main arterial road. This involves two simultaneous interventions: a dynamic tolling system expected to reduce peak-hour vehicle demand by 31%, and the replacement of the main traffic light with a 'green wave' priority junction to improve flow. What is the percentage change in average vehicle idling time? How much do total $CO_2$ emissions change, and is this reduction considered minor (less than 25%), significant (25% to 50%), or transformative (over 50%)? Does this project deliver on its promise to significantly enhance both traffic efficiency and environmental sustainability?

**Reference Answer**: The 'Smart Corridor' project demonstrates an exceptional positive impact on both traffic flow and environmental quality. Average vehicle idling time is dramatically reduced by 93.1%, signifying a near elimination of stop-and-go congestion and substantial operational efficiencies. Furthermore, the initiative achieves a significant environmental improvement with a 49.67% decrease in total $CO_2$ emissions, falling squarely within the defined 'significant' reduction category. These combined outcomes confirm that the project is highly successful in delivering on its promise of enhancing urban mobility and fostering greater environmental sustainability.

**Reference Answer Claims**:

- The 'Smart Corridor' project reduces average vehicle idling time by 93.1%.
- The 'Smart Corridor' project decreases total $CO_2$ emissions by 49.67%.
- The reduction in $CO_2$ emissions falls within the 'significant' reduction category.
- The 'Smart Corridor' project nearly eliminates stop-and-go congestion.
- The 'Smart Corridor' project enhances urban mobility.
- The 'Smart Corridor' project fosters greater environmental sustainability.

### A.3.4  DATASET STATISTICS

The statistics of two benchmark datasets are summarized in Table 2 as below:

| Benchmark | # Questions | Avg. Answer Length (words) | Avg. # Claims | Avg. # Quant. Claims | Avg. # Qual. Claims | Avg. Tool Param. Count |
|---|---|---|---|---|---|---|
| Climate | 1000 | 55.8 | 3.6 | 1.1 | 2.5 | 6.3 |
| Epidemiology | 1000 | 82.3 | 5.3 | 3.2 | 2.1 | 6.0 |
| Urban | 200 | 112.5 | 7.8 | 2.0 | 5.8 | 6.2 |

Table 2: Statistical overview of the generated benchmarks. The table compares metrics on answer length, the average number of decomposed claims (total, quantitative, and qualitative), and the average used parameter count of the simulators.

### A.4  ADDITIONAL EXPERIMENTAL RESULTS

#### A.4.1  BASELINE COMPARISON PRECISION AND RECALL

Here we provide additional experimental results, Table 3 shows the precision and recall values corresponding to the F1 scores reported in Table 1.

| Benchmark | Metric | Method | GPT-4o | | | Claude 3.5 | | |
|---|---|---|---|---|---|---|---|---|
| | | | 15% | 25% | 45% | 15% | 25% | 45% |
| Climate | Precision | Random | 0.667 | 0.6777 | 0.7043 | 0.7364 | 0.7402 | 0.7496 |
| | | Verbalized | 0.6386 | 0.6569 | 0.691 | 0.6871 | 0.6958 | 0.722 |
| | | Uncertainty | 0.6901 | 0.7113 | 0.7433 | 0.7433 | 0.7544 | 0.7926 |
| | | UE+SBA (Ours) | **0.7025** | **0.7323** | **0.7719** | **0.7513** | **0.7746** | **0.8238** |
| | Recall | Random | 0.67 | 0.6811 | 0.7036 | 0.7283 | 0.7437 | 0.7525 |
| | | Verbalized | 0.6558 | 0.6261 | **0.8046** | 0.6656 | 0.7406 | **0.9075** |
| | | Uncertainty | 0.6881 | 0.7109 | 0.7448 | 0.7372 | 0.7505 | 0.7926 |
| | | UE+SBA (Ours) | **0.7014** | **0.7297** | 0.7731 | **0.751** | **0.779** | 0.8246 |
| | F1 Score | Random | 0.6685 | 0.6794 | 0.7039 | 0.7323 | 0.7419 | 0.7511 |
| | | Verbalized | 0.6471 | 0.6411 | 0.7435 | 0.6762 | 0.7175 | 0.8042 |
| | | Uncertainty | 0.6891 | 0.7111 | 0.744 | 0.7402 | 0.7524 | 0.7926 |
| | | UE+SBA (Ours) | **0.702** | **0.731** | **0.7725** | **0.7511** | **0.7768** | **0.8242** |
| Epidemiology | Precision | Random | 0.591 | 0.6128 | 0.658 | 0.6945 | 0.7057 | 0.7353 |
| | | Verbalized | 0.561 | 0.5891 | 0.7021 | 0.6806 | 0.7018 | 0.7565 |
| | | Uncertainty | 0.6173 | 0.647 | 0.7139 | 0.7138 | 0.7387 | 0.7863 |
| | | UE+SBA (Ours) | **0.6416** | **0.6952** | **0.8131** | **0.7233** | **0.7579** | **0.82** |
| | Recall | Random | 0.5886 | 0.6185 | 0.6582 | 0.6977 | 0.7026 | 0.7395 |
| | | Verbalized | 0.5735 | 0.6589 | 0.6834 | 0.6866 | 0.7937 | 0.7065 |
| | | Uncertainty | 0.6164 | 0.6484 | 0.7126 | 0.7027 | 0.7393 | 0.7854 |
| | | UE+SBA (Ours) | **0.6445** | **0.6962** | **0.8178** | **0.7229** | 0.7608 | **0.8214** |
| | F1 Score | Random | 0.5898 | 0.6157 | 0.6581 | 0.6961 | 0.7042 | 0.7374 |
| | | Verbalized | 0.5672 | 0.6221 | 0.6926 | 0.6836 | 0.7449 | 0.7307 |
| | | Uncertainty | 0.6168 | 0.6477 | 0.7132 | 0.7082 | 0.739 | 0.7859 |
| | | UE+SBA (Ours) | **0.6431** | **0.6957** | **0.8155** | **0.7231** | **0.7594** | **0.8207** |
| Urban | Precision | Random | 0.7913 | 0.7969 | 0.8085 | 0.6776 | 0.6996 | 0.7493 |
| | | Verbalized | 0.7777 | 0.7859 | 0.8022 | 0.6863 | 0.7105 | 0.7545 |
| | | Uncertainty | 0.7949 | 0.8013 | 0.8210 | 0.6857 | 0.7063 | 0.7580 |
| | | UE+SBA (Ours) | **0.8128** | **0.8231** | **0.8344** | **0.7163** | **0.7624** | **0.8218** |
| | Recall | Random | 0.7925 | 0.7990 | 0.8124 | 0.6826 | 0.7065 | 0.7557 |
| | | Verbalized | 0.7778 | 0.7859 | 0.8022 | 0.6864 | 0.7105 | 0.7546 |
| | | Uncertainty | 0.7949 | 0.7973 | 0.8092 | 0.6821 | 0.7102 | 0.7497 |
| | | UE+SBA (Ours) | **0.8028** | **0.8216** | **0.8236** | **0.7182** | **0.7602** | **0.8225** |
| | F1 Score | Random | 0.7919 | 0.7980 | 0.8105 | 0.6801 | 0.7030 | 0.7525 |
| | | Verbalized | 0.7777 | 0.7859 | 0.8022 | 0.6863 | 0.7105 | 0.7545 |
| | | Uncertainty | 0.7949 | 0.7993 | 0.8151 | 0.6839 | 0.7082 | 0.7538 |
| | | UE+SBA (Ours) | **0.8078** | **0.8224** | **0.8290** | **0.7173** | **0.7613** | **0.8222** |

Table 3: Additional results of claim-level generation methods. The table includes Precision, Recall, and F1 Score for selection thresholds of 15%, 25%, and 45%, where the threshold is selected by letting the recall close to the precision

### A.4.2 DETAILED PERFORMANCE COMPARISON WITH CONFIDENCE INTERVALS

Table 4 presents the complete performance metrics (F1 Score, AUPR, AUROC) with 95% confidence intervals across two benchmarks (Climate, Epidemiology), confirming the statistical significance of SimulRAG's improvements.

### A.4.3 PR CURVES AMONG UE METHODS

The following precision-recall curves compare our UE+SBA method to baseline UE approaches on the Climate and Epidemiology datasets. At retrieval-augmentation-generation (RAG) rates of 25% and 45%, our results show that UE + SBA at a rate of 45% reduced half of RAG updates and achieves performance comparable to an exhaustive strategy that augments all claims.

| Benchmark | Metric | Method | GPT-4o | | | Claude 3.5 | | |
|---|---|---|---|---|---|---|---|---|
| | | | 15% | 25% | 45% | 15% | 25% | 45% |
| Climate | F1 Score | Random | $0.669 \pm 0.05$ | $0.679 \pm 0.05$ | $0.704 \pm 0.04$ | $0.732 \pm 0.03$ | $0.742 \pm 0.05$ | $0.751 \pm 0.05$ |
| | | Uncertainty | $0.689 \pm 0.02$ | $0.712 \pm 0.02$ | $0.744 \pm 0.03$ | $0.740 \pm 0.04$ | $0.753 \pm 0.03$ | $0.793 \pm 0.05$ |
| | | UE+SBA | $\mathbf{0.702 \pm 0.03}$ | $\mathbf{0.731 \pm 0.00}$ | $\mathbf{0.773 \pm 0.02}$ | $\mathbf{0.751 \pm 0.05}$ | $\mathbf{0.777 \pm 0.05}$ | $\mathbf{0.824 \pm 0.04}$ |
| | AUPR | Random | $0.714 \pm 0.02$ | $0.738 \pm 0.02$ | $0.759 \pm 0.03$ | $0.788 \pm 0.03$ | $0.793 \pm 0.02$ | $0.805 \pm 0.04$ |
| | | Uncertainty | $0.740 \pm 0.03$ | $0.754 \pm 0.01$ | $0.771 \pm 0.05$ | $0.778 \pm 0.06$ | $0.784 \pm 0.06$ | $0.809 \pm 0.04$ |
| | | UE+SBA | $\mathbf{0.746 \pm 0.02}$ | $\mathbf{0.763 \pm 0.03}$ | $\mathbf{0.787 \pm 0.02}$ | $\mathbf{0.788 \pm 0.04}$ | $\mathbf{0.804 \pm 0.02}$ | $\mathbf{0.830 \pm 0.04}$ |
| | AUROC | Random | $0.601 \pm 0.06$ | $0.627 \pm 0.06$ | $0.662 \pm 0.05$ | $0.667 \pm 0.02$ | $0.679 \pm 0.02$ | $0.703 \pm 0.04$ |
| | | Uncertainty | $0.642 \pm 0.02$ | $0.667 \pm 0.03$ | $0.704 \pm 0.02$ | $0.683 \pm 0.05$ | $0.703 \pm 0.05$ | $0.745 \pm 0.03$ |
| | | UE+SBA | $\mathbf{0.653 \pm 0.04}$ | $\mathbf{0.685 \pm 0.01}$ | $\mathbf{0.725 \pm 0.02}$ | $\mathbf{0.700 \pm 0.03}$ | $\mathbf{0.729 \pm 0.02}$ | $\mathbf{0.771 \pm 0.03}$ |
| Epidemiology | F1 Score | Random | $0.590 \pm 0.06$ | $0.616 \pm 0.07$ | $0.658 \pm 0.06$ | $0.696 \pm 0.04$ | $0.704 \pm 0.04$ | $0.737 \pm 0.03$ |
| | | Uncertainty | $0.617 \pm 0.09$ | $0.647 \pm 0.04$ | $0.714 \pm 0.04$ | $0.708 \pm 0.07$ | $0.737 \pm 0.05$ | $0.785 \pm 0.03$ |
| | | UE+SBA | $\mathbf{0.643 \pm 0.07}$ | $\mathbf{0.696 \pm 0.04}$ | $\mathbf{0.816 \pm 0.04}$ | $\mathbf{0.723 \pm 0.06}$ | $\mathbf{0.759 \pm 0.05}$ | $\mathbf{0.821 \pm 0.04}$ |
| | AUPR | Random | $0.673 \pm 0.03$ | $0.705 \pm 0.03$ | $0.756 \pm 0.06$ | $0.743 \pm 0.04$ | $0.769 \pm 0.04$ | $0.810 \pm 0.04$ |
| | | Uncertainty | $0.695 \pm 0.06$ | $0.726 \pm 0.05$ | $0.787 \pm 0.06$ | $0.755 \pm 0.08$ | $0.784 \pm 0.03$ | $0.825 \pm 0.02$ |
| | | UE+SBA | $\mathbf{0.724 \pm 0.06}$ | $\mathbf{0.775 \pm 0.07}$ | $\mathbf{0.842 \pm 0.05}$ | $\mathbf{0.774 \pm 0.06}$ | $\mathbf{0.811 \pm 0.03}$ | $\mathbf{0.859 \pm 0.02}$ |
| | AUROC | Random | $0.645 \pm 0.02$ | $0.675 \pm 0.02$ | $0.730 \pm 0.05$ | $0.572 \pm 0.04$ | $0.605 \pm 0.04$ | $0.672 \pm 0.04$ |
| | | Uncertainty | $0.691 \pm 0.04$ | $0.735 \pm 0.03$ | $0.808 \pm 0.04$ | $0.609 \pm 0.10$ | $0.657 \pm 0.06$ | $0.738 \pm 0.02$ |
| | | UE+SBA | $\mathbf{0.731 \pm 0.04}$ | $\mathbf{0.791 \pm 0.05}$ | $\mathbf{0.863 \pm 0.03}$ | $\mathbf{0.634 \pm 0.08}$ | $\mathbf{0.697 \pm 0.05}$ | $\mathbf{0.792 \pm 0.05}$ |

Table 4: Detailed performance comparison (F1, AUPR, AUROC) with 95% Confidence Intervals (CI) across all benchmarks. Verbalized baseline is omitted for brevity but follows similar trends.

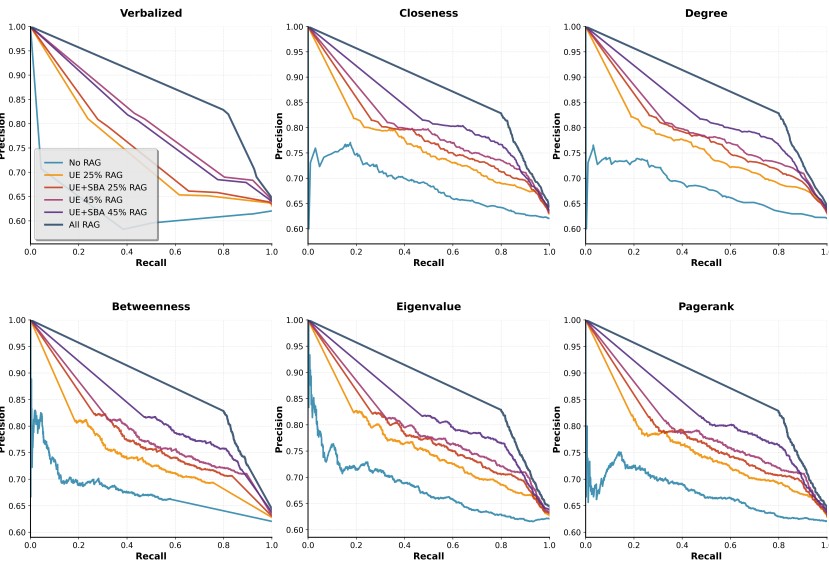

Figure 7: Precision-Recall Performance between Uncertainty, UE+SBA method across different six uncertainty estimation methods on Climate dataset.

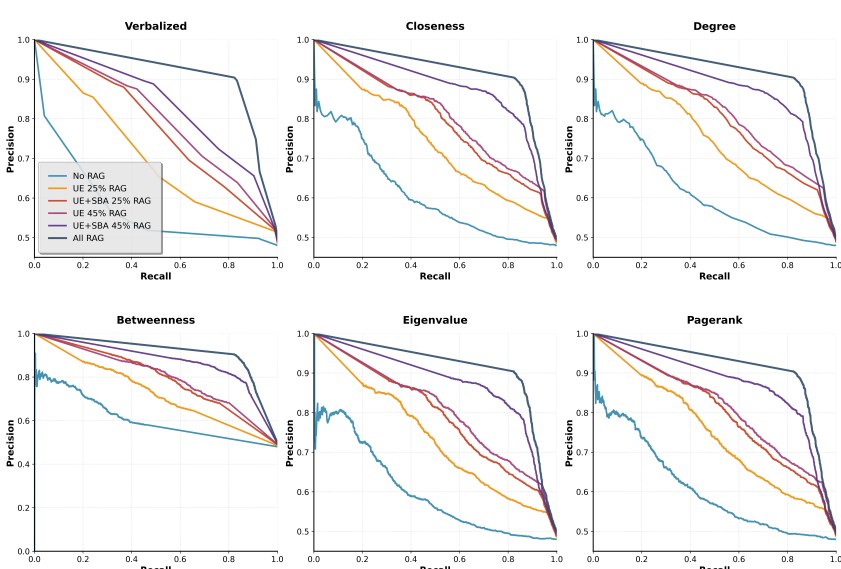

Figure 8: Precision-Recall Performance between Uncertainty, UE+SBA method across different six uncertainty estimation methods on Epidemiology dataset.

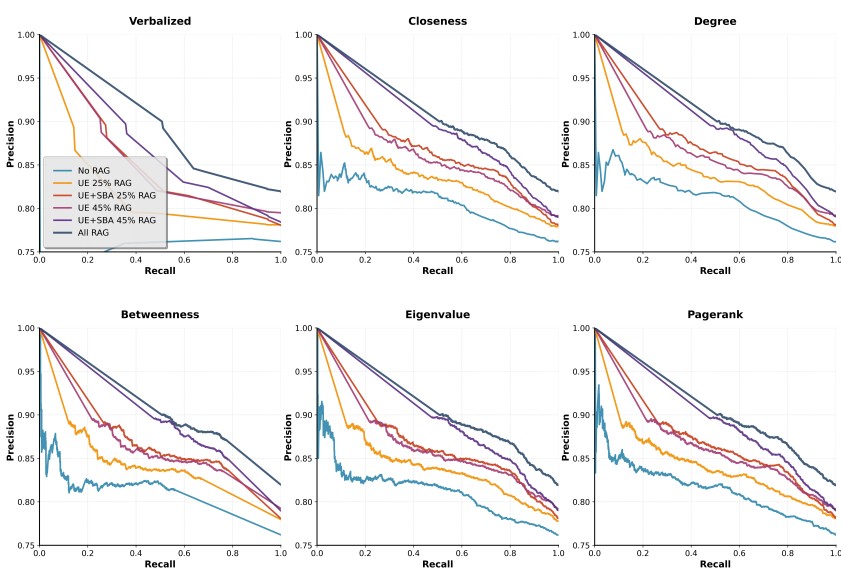

Figure 9: Precision-Recall Performance between Uncertainty, UE+SBA method across different six uncertainty estimation methods on Urban Planning dataset.

