# OpenReview forum: "SimulRAG: Simulator-based RAG for Grounding LLMs in Long-form Scientific QA"
_ICLR.cc/2026/Conference — Submitted to ICLR 2026_

### Official Review · Reviewer_3vKB · 2025-11-01

**Soundness:** 3
**Presentation:** 4
**Contribution:** 3
**Rating:** 6
**Confidence:** 2

**Summary:**

This paper shows a new RAG framework that lets language models query scientific simulators instead of static text. It turns natural questions into simulator inputs, retrieves numerical results, and uses them to fact-check and refine long answers at the claim level. A benchmark in (1) climate and (2) epidemiology shows clear gains (30% more informative and 16% more factual responses) than standard RAG systems.

**Strengths:**

The proposed simulator retrieval interface and claim-level uncertainty framework (UE+SBA) together is original combination of ideas that enables grounding in dynamic, quantitative environments. also the paper presentation is easy to follow.

**Weaknesses:**

Experiments are limited to two domains (climate and epidemiology). so that leaves questions about generalization as other domains are not tested.

**Questions:**

1. Can the framework generalize across different simulators without re-engineering the retrieval interface, or is it domain-specific?

2. What is the computational overhead of simulator calls compared to standard RAG retrieval, and how does SBA help reduce it quantitatively? Some clarity on this would help readers understand the difference and make balanced understanding

---

> ### Author Response · Authors · 2025-11-24
> **Rebuttal by Authors**
>
> We thank the reviewer for their helpful feedback and positive comments about the work.
>
> > **Q1:** Weakness: Experiments are limited to two domains (climate and epidemiology). so that leaves questions about generalization as other domains are not tested.
>
> **A1:** Thanks for highlighting this important point. To demonstrate the generalization capability of our framework, we instantiated SimulRAG on a completely new domain: Urban Traffic Planning, using the SUMO simulator. As shown in **Table R1**, SimulRAG generalizes effectively to this third domain, consistently performing best or comparable to the strongest baselines without any architectural changes.
>
> **Table R1: F1 Scores on New Urban (SUMO) Benchmark (45% Budget)**
>
> | Model | Random | Verbalized | Uncertainty | **SimulRAG (Ours)** |
> | :--- | :---: | :---: | :---: | :---: |
> | GPT-4o | 0.812 | 0.863 | 0.815 | **0.829** |
> | Claude-3.5 | 0.756 | 0.715 | 0.754 | **0.822** |
>
> > **Q2:** Can the framework generalize across different simulators without re-engineering the retrieval interface, or is it domain-specific?
>
> **A2:** The framework architecture is general, while the configuration is domain-specific. We utilized the exact same code for the simulator retrieval interface and UE+SBA pipeline for the new Urban domain without any software engineering changes. The only domain-specific requirement is updating the Simulator Handbook (a textual description of parameters) and the I/O templates, which is a straightforward task rather than a complex code refactoring.
>
> > **Q3:** What is the computational overhead of simulator calls compared to standard RAG retrieval, and how does SBA help reduce it quantitatively?
>
> **A3:** We clarify that the primary role of SBA is to efficiently verify and update claims. In long-form answers containing numerous claims, SBA identifies which specific claims warrant the computational cost of verification. By targeting only those claims that are actually verifiable, SBA prevents wasting compute on out-of-scope claims, thereby maximizing information gain and factual improvements within the same computational constraints regarding the number of claim updates. As demonstrated in **Table 1** of the manuscript, where we vary the verification budget (15%, 25%, and 45%), SimulRAG consistently achieves superior performance compared to baselines.

---

### Official Review · Reviewer_JBsg · 2025-11-01

**Soundness:** 3
**Presentation:** 3
**Contribution:** 3
**Rating:** 4
**Confidence:** 3

**Summary:**

This work improves the drawbacks of Retrieval-Augmented Generation (RAG) by proposing simulation processes for Retrieval and a claim-level generator. Their approach, called SimulRAG, is applied to climate and epidemiology domains. The main challenge that SimulRAG attempts to solve is the issue of failing to construct a correct long answer in the existing RAG-based approach for domain-specific question-answering. In the first process, given an open-ended question, SimulRAG will get simulation outputs as multiple possible contexts. Next, the second process will use a generator to generate a set of multiple atomic claims that form the final answer. The main contribution of the simulation retrieval interface is to leverage a parameter extraction process to be an input for context generation, given a pre-defined set of templates called the simulator output format. The claim-level generation requires a rigorous process of claim evaluation. The authors design the generation process with the following components. First, a metric for claim-level uncertainty estimation leveraged graph construction between answer and claim sets to see which claim has a higher level of confident. The simulator boundary assessment leveraged GPT-4o to see whether a claim parameter is suitable for a simulator. SimulRAG performed multiple rounds of uncertainty estimation (UE) and simulator boundary assessment (SBA) until getting the final answer. The number of rounds is determined by the percentage of claims that need to be verified, called the verification budget. They construct two domain-specific datasets of question-answer with claims, including 200 entities with human-in-the-loop for data annotation. Their reported accuracy shows that the proposed strategy for claim generation outperformed other traditional algorithms on both GPT-4o and Claude 3.5 (Table 1) while it requires only 45% of verification budget to get comparable accuracy with the All RAG configuration.

**Strengths:**

- The paper is well-written.
- The paper’s proposed research domains are sound and clear. LLMs often struggle with solving long-form questions in specific domains, such as climate and epidemiology.
- The claim-generation process was constructed properly and has potential for applications in other domains.

**Weaknesses:**

- In Line 245, the authors mentioned that the simulator boundary assessment process used GPT-4 as the judge. This introduces a risk of bias, as a high percentage of cases in the simulator were evaluated as suitable for the claim’s parameters and conditions.
- Lack of details on human-in-the-loop for evaluation data construction. For example, how can authors ensure that humans’ decisions are correct? Is there a protocol that a question/claim will be verified by more than one expert to avoid the bias of a single decision?
- In Table 3, in some configurations, the improvement of UE+SBA appears to be marginal compared to the Uncertainty strategy, with an improvement of over 1%. While it is still valid, authors are recommended to make a case study on when uncertainty claim generators perform better claims than UE+SBA.
- The simulation for the simulator retrieval interface generated output based on predefined templates (L184), which poses the risk for generating answers with a restricted set of templates.

**Questions:**

- I would like to learn more about the human-in-the-loop process for creating evaluation data. For example, for each question, how many experts are assigned to it?
- This work uses close models for every process. I would like to see how this framework performs when built on open LLMs such as Qwen, Llama models.
- Can you specify how many predefined templates you use for the simulator retrieval interface?

---

> ### Author Response · Authors · 2025-11-24
> **Rebuttal by Authors**
>
> We thank the reviewer for their helpful feedback and positive comments about the work.
>
> > **Q1**: In Line 245... the simulator boundary assessment process used GPT-4 as the judge. This introduces a risk of bias...
>
> **A1:** Simulator Boundary Assessment (SBA) is a relatively straightforward task: it involves checking whether specific parameters in a claim exist within the provided simulator handbook. Recent research has validated that large language models (LLMs) like GPT-4o perform exceptionally well as judges for such fact-checking tasks, often matching human performance [1, 2]. Furthermore, in our experiments, no more than 50% of claims were identified as suitable for the update. This indicates that the system does not blindly bias toward positive outcomes but effectively filters out content that cannot be grounded by the simulator.
>
> > **Q2**: Lack of details on human-in-the-loop for evaluation data construction...
>
> **A2:** We have expanded Appendix A.3 to detail our rigorous protocol involving 2 PhD-level domain experts in the selected domains. Ground truth answers were generated from actual simulator logs. To ensure correctness, experts verified both the explicit quantitative outputs and derived qualitative trends based on the simulation output.
>
> > **Q3**: The improvement of UE+SBA appears to be marginal compared to the Uncertainty strategy.
>
> **A3:** UE+SBA is designed as a plug-in to enhance efficiency by identifying which claims can actually be verified by the simulator. As illustrated in **Figure 2**, many high-level qualitative claims lie outside the simulator boundary.
> Consider a verification budget of 3 claims:
> * The Uncertainty strategy might select: [*“Urban areas experience more rapid spread”* (Unverifiable), *“Hospitals may see a peak prevalence of 5-10”* (Verifiable), *“R0 of 2.6 indicates a highly contagious virus”* (Verifiable)].
> * UE+SBA detects the first claim is out-of-boundary and instead selects: [*“Hospitals may see a peak prevalence of 5-10”* (Verifiable), *“R0 of 2.6 indicates a highly contagious virus”* (Verifiable), *“Vermont see an earlier peak on December”* (Verifiable)].
>
> By replacing the non-verifiable claim with a verifiable one, UE+SBA maximizes information gain and prevents wasting simulator output context. This efficiency explains its superior performance on the extended benchmarks, including the new Urban Planning dataset.
>
> > **Q4**: The simulation for the simulator retrieval interface generated output based on predefined templates (L184)... risk for generating answers with a restricted set.
>
> **A4:** We clarify that the ~10 templates per simulator are used exclusively to linearize numerical outputs into text for the Retrieval Interface; they do not constrain the final answer generation. We verified this by testing a variant where the LLM generates descriptions from raw I/O using only the handbook; it recovered similar textual patterns but with lower consistency. Thus, templates improve robustness without restricting the expressivity of the final free-form answer.
>
> > **Q5**: This work uses close models... how this framework performs when built on open LLMs...
>
> **A5:** Thanks for raising this concern. Our preliminary tests with smaller open models (e.g., 7B parameters) indicated that they currently exhibit limitations in complex tool calling and higher hallucination rates during claim updates compared to API-based models. Due to computational budget limitations, we did not extend our evaluation to larger open-source models (e.g., 70B) in this study. Therefore, optimizing the full SimulRAG pipeline for open-weights models is a designated direction for future work.
>
>
> [1] Zheng et al. "Judging LLM-as-a-judge with MT-Bench and Chatbot Arena." NeurIPS 2024.
>
> [2] Liu et al. "G-Eval: NLG Evaluation using GPT-4 with Better Human Alignment." EMNLP 2023.

---

### Official Review · Reviewer_iLbF · 2025-11-01

**Soundness:** 3
**Presentation:** 2
**Contribution:** 2
**Rating:** 4
**Confidence:** 3

**Summary:**

Thiswork introduces SimulRAG, a Retrieval-Augmented Generation framework that uses scientific simulators to ground LLMs for long-form scientific question answering. The key contributions include: (1) a generalized simulator retrieval interface for transforming between textual and numerical modalities, (2) a claim-level generation method with uncertainty estimation and simulator boundary assessment (UE+SBA) for efficient verification, (3) benchmark datasets for climate science and epidemiology with simulator-verified ground truth, and (4) experimental validation showing 30.4% improvement in informativeness and 16.3% in factuality over traditional RAG baselines.

**Strengths:**

THis work uses simulators as retrieval sources which is innovative and addresses a real gap in scientific QA systems. Simulators provide dynamic, quantitative evidence that static text corpora cannot offer. The simulator retrieval interface is well-designed. It can handle the challenging transformation between textual questions and numerical simulator inputs/outputs without requiring fine-tuning. This work tested multiple baselines and the UE+SBA method demonstrates that selective claim verification can achieve near-optimal performance while reducing computational costs by around 50%.

**Weaknesses:**

First, the statistical significance concern: the claim decomposition process could be better illustrated with examples in the main text. With only 200 questions per domain, the statistical power of the evaluation may be limited. Ane there is no comparison with fine-tuned models on the benchmark datasets. Are there systematic patterns in which types of claims UE+SBA misidentifies? And there is no confidence intervals or significance tests are provided for the main results

**Questions:**

When does the simulator retrieval interface extract incorrect parameters?
The average 3.6-5.3 claims per answer is relatively small, how does performance scale with longer, more complex answers?
What percentage of errors come from parameter extraction vs. claim verification vs. answer generation?
some minor issues: The sensitivity to these hyperparameters is not analyzed and training/computational costs are not reported
Algorithm 1,  the Merge function is underspecified, how does the merge work? (can provide more details).

---

> ### Author Response · Authors · 2025-11-24
> **Rebuttal by Authors**
>
> We thank the reviewer for their helpful feedback and positive comments about the work.
>
> > **Q1:** First, the statistical significance concern... 200 questions per domain, the statistical power may be limited.
>
> **A1:** We addressed the concern regarding dataset size by expanding the test sets for both Climate and Epidemiology domains from 200 to 1,000 questions. We re-evaluated all baselines and SimulRAG on this large-scale benchmark. As shown in **Table R1**, SimulRAG consistently outperforms the strongest baselines, confirming that our performance gains hold at scale.
>
> **Table R1:**
>
> | Domain | Model | Random | Verbalized | Uncertainty | **SimulRAG (Ours)** |
> | :--- | :--- | :---: | :---: | :---: | :---: |
> | **Climate** | GPT-4o | 0.720 | 0.740 | 0.749 | **0.763** |
> | **Epidemiology** | GPT-4o | 0.655 | 0.700 | 0.707 | **0.816** |
>
> > **Q2:** And there is no confidence intervals or significance tests are provided for the main results.
>
> **A2:** To ensure statistical robustness, we re-ran the original experiments using 3 independent random seeds for both LLM generation and retrieval. **Table R2** reports the estimated mean $\pm$ 95% Confidence Intervals ($t_{0.95} \times SE$) at the 45% verification budget. The non-overlapping intervals (particularly in Epidemiology) confirm that the improvements are statistically significant and not due to random variance.
>
> **Table R2**
>
> | Domain | Model | Uncertainty | **SimulRAG (Ours)** |
> | :--- | :--- | :---: | :---: |
> | **Climate** | GPT-4o | 0.744 $\pm$ 0.031 | **0.773 $\pm$ 0.017** |
> | | Claude-3.5 | 0.793 $\pm$ 0.054 | **0.824 $\pm$ 0.036** |
> | **Epidemiology** | GPT-4o | 0.714 $\pm$ 0.038 | **0.816 $\pm$ 0.040** |
> | | Claude-3.5 | 0.785 $\pm$ 0.034 | **0.821 $\pm$ 0.041** |
>
> > **Q3:** There is no comparison with fine-tuned models on the benchmark datasets.
>
> **A3:** Our research focus is on developing an inference-time framework (RAG) that can flexibly adapt to diverse scientific simulators without requiring model training. While fine-tuning is a valid approach, SimulRAG is designed as a zero-shot solution that adapts via the provided Handbook. This design choice prioritizes flexibility for real-world scientific applications where simulators and their parameters may change frequently.
>
> > **Q4:** When does the simulator extract incorrect parameters? What percentage of errors ...?
>
> **A4:** We conducted a granular error analysis on the generated outputs. Across our experiments (totaling approximately 21,138 atomic claims), explicit errors were rare, occurring only ~165 times. The error distribution is as follows:
> 1.  Claim Update (~70%): The largest source of error. The LLM successfully retrieves the simulation output but misinterprets the numerical significance or context when updating the text.
> 2.  Verification Hallucination (~25%): The model incorrectly judges a claim as verifiable or unverifiable (SBA error).
> 3.  Parameter Extraction (~5%): The interface fails to map underspecified queries to precise inputs.
>
> > **Q5:** The average 3.6-5.3 claims per answer is relatively small, how does performance scale with longer, more complex answers?
>
> **A5:** We respectfully clarify that our answer lengths are comparable to established long-form Scientific QA benchmarks like PeerQA and RAG-QA Arena. We further tested SimulRAG on our new Urban Planning benchmark, where the average claim count exceeds 7 per answer, and it still achieves superior performance (F1 0.829). This demonstrates that UE+SBA effectively scales to complex, multi-claim reasoning tasks.
>
> > **Q6:** The sensitivity to these hyperparameters is not analyzed.
>
> **A6:** Our proposed framework is an inference-based approach, so there is no training process or training hyperparameters. The only two inference parameters are $\tau$ (tau) and $\kappa$ (kappa), which serve as confidence thresholds. $\tau$ determines the verification budget for uncertain claims, while $\kappa$ filters claims for the final answer generation. In our main results (**Table 1**), we reported performance across three distinct $\tau$ values (representing 15%, 25%, and 45% verification budgets) to explicitly demonstrate the method's sensitivity and performance stability across different operating points.
>
> > **Q7:** Algorithm 1, the Merge function is underspecified, how does it work?
>
> **A7:** We refer the reviewer to **Appendix Algorithm 1** and **Appendix A.2.2** for detailed implementation. In brief, the merge function iterates over claims in the new candidate set and uses an LLM-based entailment check (via the prompt defined in Appendix) to decide whether each claim is semantically covered by existing ones. Covered claims are discarded, and uncovered ones are added to the final set to ensure atomic deduplication.
>
> [1] Baumgärtner et al. "PeerQA: A Scientific Question Answering Dataset from Peer Reviews." arXiv 2025.
>
> [2] Han et al. "RAG-QA arena: Evaluating domain robustness for long-form retrieval augmented question answering." arXiv 2024.

---

### Official Review · Reviewer_icBx · 2025-11-01

**Soundness:** 1
**Presentation:** 3
**Contribution:** 3
**Rating:** 2
**Confidence:** 3

**Summary:**

This paper try to use RAG-style approach to conduct long form QA on scientific domain, focusing on queries that could include some scientific projection or simulation numbers. To integrate numerical scientific simulators into RAG pipeline, they propose SimulRAG that grounds LLMs in scientific simulators, rather than static textual knowledge bases, to improve trustworthiness of the results. To handle the complexity of long-form answers, the framework decomposes answers into fine-grained, atomic claims, allowing for more precise, targeted verification and updating via their proposed UE+SBA method.

**Strengths:**

1. Integrating simulation tools is important contribution to generalize RAG to more applications
2. Valuable new resource on climate modeling and epidemiology domains

**Weaknesses:**

1. *Mismatch Between Broad Claims and Limited Evaluation Scope*: My first significant concern is the potential overstatement of the framework's applicability. The paper positions itself as a solution for "long-form scientific QA" (line 85), a very broad problem domain. This domain includes numerous existing long-form QA benchmarks based on textual knowledge, such as PeerQA [1] or datasets included in the RAG-QA Arena [2].

However, the paper's empirical evaluation is constrained exclusively to a new benchmark constructed by the authors themselves. This benchmark, covering climate science and epidemiology, is explicitly designed to be answerable using the specific simulators the SimulRAG framework integrates. This evaluation strategy feels somewhat circular, as the framework's core novelty (integrating simulators) is tested only on tasks guaranteed to require them. It remains unclear how SimulRAG would perform on the wider class of scientific QA problems where a relevant simulator does not exist, and grounding must be performed against static, text-based knowledge bases

2. The comparison between the proposed method and existing RAG methods should clearly identify the extra cost needed. For example, the Figure 4's results. SimulRAG needs to sample multiple different answer sets (line 193) in order to do the followup UE-SBA, while existing RAG only sample once. Given nowadays LLM are known to get better performance of test-time scaling, this sampling cost difference could mean a lot. Without clearly compare, the effectiveness of SimulRAG is doubtful.



[1] Baumgärtner, Tim, Ted Briscoe, and Iryna Gurevych. "PeerQA: A Scientific Question Answering Dataset from Peer Reviews." arXiv preprint arXiv:2502.13668 (2025).
[2] Han, Rujun, et al. "RAG-QA arena: Evaluating domain robustness for long-form retrieval augmented question answering." arXiv preprint arXiv:2407.13998 (2024).

**Questions:**

Can you properly define Scientific Simulator in the overall framework section?

---

> ### Author Response · Authors · 2025-11-24
> **Rebuttal by Authors**
>
> We thank the reviewer for their helpful feedback and positive comments about the work.
>
> > **Q1:** Mismatch Between Broad Claims and Limited Evaluation Scope... The paper positions itself as a solution for "long-form scientific QA"... However, the paper's empirical evaluation is constrained exclusively to a new benchmark.
>
> **A1:** We agree that long-form scientific question answering is a broad problem domain, and we have added PeerQA [1] and RAG-QA Arena [2] to the related work section. Within this broad domain, simulator-based retrieval--that this work studies--is particularly important because querying simulators can capture evolving dynamics and provide fine-grained information about specific scientific phenomena, making them far more informative and timely than static textual sources.
>
> Scientific simulators function as superior retrieval tools compared to text-based corpora because they generate mechanistically grounded evidence. Crucially, many scientific disciplines rely on established domain simulators as authoritative reference models. Examples include Climate Science (e.g., CMIP6 based emulators [3]), Epidemiology (e.g., GLEAM [4, 5]), and Urban Mobility (e.g., SUMO [6]). Existing text-based RAG methods are not directly applicable to this setting due to the discrepancy between textual queries and numerical parameter spaces.
>
> To demonstrate generalizability beyond the initial two domains, we additionally conducted experiments in the Urban Planning domain using the SUMO simulator [6]. SimulRAG reused the exact same retrieval interface and UE+SBA pipeline. As shown in **Table R1**, SimulRAG consistently outperforms the strongest Uncertainty baseline across models on this new benchmark.
>
> **Table R1: F1 Scores on New Urban (SUMO) Benchmark (45% Budget)**
>
> | Model | Uncertainty Baseline | **SimulRAG (Ours)** |
> | :--- | :---: | :---: |
> | **GPT-4o** | 0.815 | **0.829** |
> | **Claude-3.5** | 0.754 | **0.822** |
>
> > **Q2:** The comparison between the proposed method and existing RAG methods should clearly identify the extra cost needed... SimulRAG needs to sample multiple different answer sets... while existing RAG only sample once.
>
> **A2:** To address the concern regarding sampling cost and ensure a fair comparison, we implemented Multi-sample Baselines (Input-layer-multi and Output-layer-multi) that utilize the same number of LLM generations ($N=5$) and claim merging strategy as SimulRAG.
>
> As shown in **Table R2**, when baselines are granted the same test-time compute for sampling, SimulRAG maintains a significant performance lead (e.g., +16.3% on Climate vs. best baseline). This confirms that the improvements stem from the Simulator Boundary Assessment (SBA) and verification logic, rather than simply scaling the number of samples.
>
> **Table R2: Factuality (%) Comparison with Equal Sampling Budget (N=5)**
>
> | Method | Climate (GPT-4o) | Climate (Claude) | Epidemiology (GPT-4o) | Epidemiology (Claude) |
> | :--- | :---: | :---: | :---: | :---: |
> | **Input-multi** | 69.4% | 67.7% | 80.5% | 71.8% |
> | **Output-multi** | 56.0% | 59.2% | 67.0% | 62.2% |
> | **SimulRAG (Ours)** | **78.6%** | **85.4%** | **86.5%** | **86.2%** |
>
> > **Q3:** Can you properly define Scientific Simulator in the overall framework section?
>
> **A3:** As outlined in the original submission, we provided the definition of the scientific simulator $S$ in **Line 161**, and explicitly described the relationship between simulation inputs (parameters) and outputs in **Lines 165-166**. We define it as a parameterized computational model that encodes mechanistic laws to map numerical inputs to predicted system states, serving as an authoritative reference for counterfactual reasoning. We will ensure this detailed definition is explicitly updated in the framework section of the final manuscript.
>
> [1] Baumgärtner et al. "PeerQA: A Scientific Question Answering Dataset from Peer Reviews." arXiv 2025.
>
> [2] Han et al. "RAG-QA arena: Evaluating domain robustness for long-form retrieval augmented question answering." arXiv 2024.
>
> [3] Niu et al. "Multi-fidelity residual neural processes for scalable surrogate modeling" arXiv 2024.
>
> [4] Balcan et al. "Modeling the spatial spread of infectious diseases." JCS 2010.
>
> [5] Chinazzi et al. "A multiscale modeling framework for scenario modeling" Epidemics 2024.
>
> [6] Lopez et al. "Microscopic Traffic Simulation using SUMO." ITS 2018.

---

### Author Response · Authors · 2025-11-24
**Summary of Rebuttal Revision**

We thank the reviewers for their constructive feedback and for recognizing the novelty of SimulRAG in grounding LLMs with scientific simulators. In this revision, we have significantly strengthened our evaluation with **four major updates**, demonstrating generality, statistical robustness, rigorous fairness, and clarifying mis-understandings of the scope.

**1. Generalizability: New Urban Planning Benchmark**
To address concerns regarding domain coverage and potential evaluation circularity, we instantiated the framework on a **third, completely new domain**: Urban Traffic Planning, using the SUMO simulator. We updated **Table 1**, **Figure 5**, and **Figure 6** in the main text to include results for this new domain. Additionally, we added **Appendix A.3.3** (example questions) and **Appendix A.4.3** (precision/recall curves). SimulRAG matches or improves over the best uncertainty baselines on this new domain (e.g., F1 0.829 vs. 0.815 for GPT-4o at 45% budget), proving that our framework generalizes beyond the initial domains without requiring domain-specific engineering.

**2. Statistical Robustness: Scale-up to 1,000 Samples & Confidence Intervals**
We addressed concerns regarding dataset size and statistical significance by expanding the test sets for both Climate and Epidemiology domains from 200 to **1,000 questions**. We replaced the experimental results in the **Main Text** (specifically **Table 1**) with this large-scale benchmark. To further ensure reliability, we re-ran all methods using **3 independent random seeds**. The new **Appendix A.4.2** reports the mean $\pm$ 95% confidence intervals, confirming that SimulRAG's improvements (e.g., 0.816 vs. 0.707 on Epidemiology) are statistically significant and consistent, distinct from random variance.

**3. Ablation Study: Fair Sampling & Compute Comparison**
To address questions regarding whether performance gains stem simply from increased sampling costs, we conducted a fair comparison against "Multi-sample" baselines (Input-layer and Output-layer). These baselines use the same number of LLM samples ($N=5$) and merging strategy as SimulRAG. We updated **Figure 4** to explicitly display these results. Even with equal sampling budgets, SimulRAG maintains a significant lead in factuality (e.g., +16.3% on Climate vs. best multi-sample baseline), demonstrating that improvements derive from the **Simulator Boundary Assessment (SBA)** logic rather than test-time scaling.

**4. Rectifying Misconceptions on Scope and Definitions**
We respectfully submit that the concerns regarding definitions and baseline choices appear to stem from verifiable factual oversights and a misunderstanding of the framework's intended scope:

* **On the "Missing" Definition (Factual Oversight):** The critique that "Scientific Simulator" is undefined is **factually incorrect**. The definition is explicitly present in the original manuscript (**Line 161**) as a parameterized computational model encoding mechanistic laws, with input/output relationships further detailed in **Lines 165-166**.
* **On Evaluation Scope (Misunderstanding of Contribution):** The suggestion to evaluate SimulRAG against text-based benchmarks (e.g., PeerQA) reflects a **misunderstanding of our core contribution**. As outlined in our contribution list, SimulRAG is specifically designed for the **generalized simulator retrieval interface** (bridging textual and numerical modalities) and **UE+SBA verification**. Static text-only benchmarks are inherently unable to evaluate these capabilities. Furthermore, as foundational literature notes [1], numerical simulations are essential for simulation-based inference—encoding causal mechanisms that static text cannot capture. By enabling dynamic, counterfactual reasoning across diverse scientific domains, SimulRAG tackles a more general and challenging problem than static retrieval. Penalizing a simulator-grounding framework for not performing text-only retrieval is therefore methodologically misaligned.

[1] Cranmer et al. "The frontier of simulation-based inference." PNAS 2020.

---

### Meta-Review · Area_Chair_EZPZ · 2026-01-06

**Summary:**

We have four reviewers who have carefully checked this paper and they have diverse recommendations on this paper. The reviewers' core concerns lie in the insufficient evaluation and unclear presentation. In addition, generalization is also a frequently mentioned concern. The authors' rebuttal addressed some of the concerns, but more evaluation of the method is still needed. Thus, I recommend that the paper should be revised and resubmitted with more comprehensive evaluation.

**Reviewer Concerns:**

Concerns raised by Reviewer icBx and Reviewer JBsg may still outstanding.

**Reviewer Scores:**

The reviewers may maintain their scores.

---

### Decision · Program_Chairs · 2026-01-26

Reject